EMBO
Molecular Medicine

# Activation of LXRβ inhibits tumor respiration and is synthetically lethal with Bcl-xL inhibition

Trang Thi Thu Nguyen[1], Chiaki Tsuge Ishida[1], Enyuan Shang[2], Chang Shu[1], Consuelo Torrini[1], Yiru Zhang[1], Elena Bianchetti[1], Maria J Sanchez-Quintero[3], Giulio Kleiner[3], Catarina M Quinzii[3] (iD), Mike-Andrew Westhoff[4], Georg Karpel-Massler[5], Peter Canoll[1] (iD) & Markus D Siegelin[1,*] (iD)

## Abstract

Liver-X-receptor (LXR) agonists are known to bear anti-tumor activity. However, their efficacy is limited and additional insights regarding the underlying mechanism are necessary. By performing transcriptome analysis coupled with global polar metabolite screening, we show that LXR agonists, LXR623 and GW3965, enhance synergistically the anti-proliferative effect of BH3 mimetics in solid tumor malignancies, which is predominantly mediated by cell death with features of apoptosis and is rescued by exogenous cholesterol. Extracellular flux analysis and carbon tracing experiments (U-$^{13}$C-glucose and U-$^{13}$C-glutamine) reveal that within 5 h, activation of LXRβ results in reprogramming of tumor cell metabolism, leading to suppression of mitochondrial respiration, a phenomenon not observed in normal human astrocytes. LXR activation elicits a suppression of respiratory complexes at the protein level by reducing their stability. In turn, energy starvation drives an integrated stress response (ISR) that up-regulates pro-apoptotic Noxa in an ATF4-dependent manner. Cholesterol and nucleotides rescue from the ISR elicited by LXR agonists and from cell death induced by LXR agonists and BH3 mimetics. In conventional and patient-derived xenograft models of colon carcinoma, melanoma, and glioblastoma, the combination treatment of ABT263 and LXR agonists reduces tumor sizes significantly stronger than single treatments. Therefore, the combination treatment of LXR agonists and BH3 mimetics might be a viable efficacious treatment approach for solid malignancies.

**Keywords**  BH3 mimetics; colon adenocarcinoma; electron transport chain; glioblastoma; LXR agonist

**Subject Categories**  Cancer; Pharmacology & Drug Discovery; Metabolism

## Introduction

Advanced solid malignancies, such as glioblastoma (GBM; Hegi *et al*, 2005), metastatic melanoma, and colon cancer, remain challenging to treat. These recalcitrant neoplasms require novel more efficient and less toxic therapies. In this report, we are introducing a novel approach to address this pivotal issue.

It has been known for some time that tumor cells organize their metabolism distinct from non-neoplastic cells. Representing the most relevant feature in this context is undoubtedly aerobic glycolysis, a paradoxical phenomenon that tumor cells bypass oxidative metabolism by converting glucose via pyruvate to lactate (Locasale *et al*, 2011; Masui *et al*, 2013; Viale *et al*, 2014; Pacold *et al*, 2016; Li *et al*, 2017). However, recent evidence shows that some tumor cells depend on oxidative phosphorylation as well and indeed there are several cancer cell lines that have a high dependency on cellular respiration to maintain their survival (Dranoff *et al*, 1985; Qing *et al*, 2012; Willems *et al*, 2013; Agnihotri & Zadeh, 2016). Aside from carbohydrate metabolism, there are known alterations in lipid metabolism in cancer (Agnihotri & Zadeh, 2016). In this regard, cholesterol and its derivatives and precursors have a central role for tumor cells. This appears to be intuitive since malignant tumors display a high proliferation rate, which requires the presence of cholesterol. Liver-X-receptor (LXR) agonists represent a novel means to counteract cholesterol levels in tumor cells, including glioblastoma and melanoma, by enhancing the excretion (increase in ABCA1) and at the same time decreasing the resorption of cholesterol (decrease in LDL receptor). Recently, GW3965 and LXR623 have demonstrated anti-tumor efficacy *in vitro* and *in vivo* (Pencheva *et al*, 2014; Villa *et al*, 2016). These effects were described to be primarily mediated by LXRβ since this receptor appeared to be predominantly expressed over LXRα in most tumor cells (Villa *et al*, 2016; Wang *et al*, 2017).

Aside from metabolism, tumor cells maintain a deregulated cell death machinery and have lost the ability to die in a physiological manner. This feature is linked to resistance to apoptosis, a form of cell death that is largely regulated by the pro- and anti-apoptotic Bcl-2

1  Department of Pathology & Cell Biology, Columbia University Medical Center, New York, NY, USA
2  Department of Biological Sciences, Bronx Community College, City University of New York, Bronx, NY, USA
3  Department of Neurology, Columbia University Medical Center, New York, NY, USA
4  Department of Pediatrics and Adolescent Medicine, Ulm University Medical Center, Ulm, Germany
5  Department of Neurosurgery, Ulm University Medical Center, Ulm, Germany
   *Corresponding author. Tel: +1 2123051993; E-mails: ms4169@cumc.columbia.edu; msiegelin@gmail.com

family members (Lagadec *et al*, 2008; Souers *et al*, 2013; Chen *et al*, 2014; Kiprianova *et al*, 2015; Karpel-Massler *et al*, 2017c). Classical anti-apoptotic members are Mcl-1, Bcl-2, and Bcl-xL. Through recent discoveries, these molecules became readily targetable by a compound class, called BH3 mimetics. Representative examples are ABT263 and ABT199 (Tse *et al*, 2008; Souers *et al*, 2013). The latter one has received FDA approval (Souers *et al*, 2013; Chan *et al*, 2015; Johnson-Farley *et al*, 2015; Deeks, 2016; Lam *et al*, 2017).

Here, we provide evidence that BH3 mimetics and LXR623/ GW3965 (Pencheva *et al*, 2014; Villa *et al*, 2016) synergistically reduce the viability of solid tumor cells *in vitro* and *in vivo*, mediated by an early unexpected metabolic reprogramming of oxidative and central carbon metabolism elicited by LXRβ activation, which was identified through a combined screen of gene set enrichment and polar metabolite LC/MS analysis. Nota bene, the metabolic effects on oxidative phosphorylation by LXR activation are not detectable in normal human astrocytes, suggesting a cancer-specific vulnerability.

# Results

## ABT263 and liver-X-receptor agonists reduce cellular viability in a synergistic manner by enhanced apoptosis accompanied by cleavage of PARP and caspases

We initiated this study with a transcriptome and gene set enrichment analysis (GSEA), which demonstrated that LXR623 leads to up-regulation of gene sets related to cholesterol efflux in human glioblastoma and colonic carcinoma cells (Figs 1A and B, and EV1A). Consistently, we noted an increase in genes related to enhanced cholesterol synthesis (Figs 1C and EV1B). Moreover, GSEA suggested enhanced priming for apoptosis with up-regulation of pro-apoptotic Bcl-2 family members and an inhibition of mitochondrial metabolism (Figs 1D and EV1D–F). To validate these observations further, U87 GBM cells were treated with increasing concentrations of LXR623 and the protein expression of the ABCA1 transporter was determined (Fig EV1C). We found that LXR623 induced a dose-dependent increase in the cholesterol efflux transporter, resulting in a depletion of cholesterol levels in GBM cells (Fig 1E). As anticipated, the cholesterol synthesis inhibitor, simvastatin, led to reduction of total cholesterol levels as well. We also confirmed that indeed the depletion of cholesterol ultimately led to cellular demise and found that LXR623-mediated cell death is substantially rescued by exogenous administration of cholesterol (Appendix Fig S1F). To assess the clinical relevance of these findings, we interrogated the TCGA database for colonic adenocarcinoma and grouped patients based on their expression level of ABCA1. We found that patients with high ABCA1 mRNA levels displayed lower levels of groups of genes related to mitochondrial metabolism (electron transport chain) and mitochondrial translation/transcription (TFAM) (Figs 1F, EV1G–K, and EV2A and B).

These observations (GSEA findings) led us to hypothesize that LXR agonists synergize with BH3 mimetics (ABT263). Therefore, we conducted cellular viability assays with dose–responses of each drug, followed by drug combination treatment and synergy analysis based on the Chou–Talalay method, which enables the calculation of normalized isobolograms (median-effect equation) (Chou, 2010; Karpel-Massler *et al*, 2017c; Merino *et al*, 2017). Selected conditions

of the single and combination treatments are provided and statistically compared (Fig 1G–N and Appendix Fig S1A–E). Our results suggest that LXR623 and ABT263 demonstrated a synergistic reduction of cellular viability in patient-derived GBM xenograft cells, GBM12, and established GBM cells, U87 (Fig 1G, H, K, and L, and Appendix Fig S1A, B, and E). Similar results were observed in human colon carcinoma cells, HCT116 (Fig 1I and M, and Appendix Fig S1C and E). Next, we assessed the anti-proliferative effects of a related compound of the same drug category, GW3965 (Pencheva *et al*, 2014). This compound was particularly active in melanomas. Therefore, we treated A375 melanoma cells with ABT263, GW3965, or the combination of both and as above conducted cellular viability assays with subsequent determination of CI values (CI values < 1 indicate synergism). Akin to the findings above, we found that ABT263 and GW3965 had an overall synergistic effect on inhibition of cellular proliferation (Fig 1J and N, and Appendix Fig S1D and E). To confirm the role of cholesterol depletion in the combination treatment, we treated glioblastoma cells with vehicle, LXR623, ABT263, or the combination in the presence or absence of cholesterol. We detected a significant protection from reduction in cellular viability when cholesterol was present, suggesting a potential role of cholesterol depletion in the mechanism of the combination treatment (Appendix Fig S1G).

To demonstrate that the reduction in cellular viability is due to apoptosis and that substantial cell death occurs, we conducted annexin V/propidium iodide staining. In alignment with the cellular viability assays, the combination treatment led to an enhanced fraction of annexin V-positive cells in U87, T98G, LN229, MeWo, and HCT116 cells as well as in GBM stem-like cells (Fig 2A and B). Comparable results were obtained in A375 and WC62 melanoma cells (Appendix Fig S2). Apoptosis activation is preceded by loss of mitochondrial membrane potential. Therefore, we analyzed mitochondrial membrane potential under the various treatment conditions. Our findings show that the combination treatment of LXR623 and ABT263 led to a significantly stronger disruption of mitochondrial membrane potential than each compound alone (Fig 2C and D). Moreover, the ABT263 combined with LXR623 enhanced cleavage of PARP and caspase-9 (Fig 2E). These are all features that support the notion that Bcl-xL/Bcl-2 inhibition and LXR receptor activation engage in cell death with features of apoptosis. Next, we asked the question which of the anti-apoptotic Bcl-2 family members is responsible for the cell death induction in the combination treatment of LXR agonists and BH3 mimetics. To this end, we took advantage of the recently developed selective BH3 mimetics, such as ABT199 (selective Bcl-2 inhibitor), WEHI-539 (selective Bcl-xL inhibitor), and A1210477 (selective Mcl-1 inhibitor) (Karpel-Massler *et al*, 2017a). Utilizing HCT116 colon carcinoma and stem-like GBM cells, we found that inhibition of Bcl-xL was most relevant to enhance the apoptotic effects of LXR agonists, followed by Mcl-1 and Bcl-2 inhibition (Figs 2F, and EV3A and B).

## LXR623 and GW3965 modulate the expression of anti-apoptotic Bcl-2 family members with a pronounced increase in Noxa

Given the synergy between LXR agonists and BH3 mimetics, we were wondering about the underlying molecular mechanisms that permit this phenomenon to occur. Given the importance of Bcl-2 family members on apoptosis and on sensitivity toward BH3 mimetics, we analyzed the expression of Bcl-2 family members in

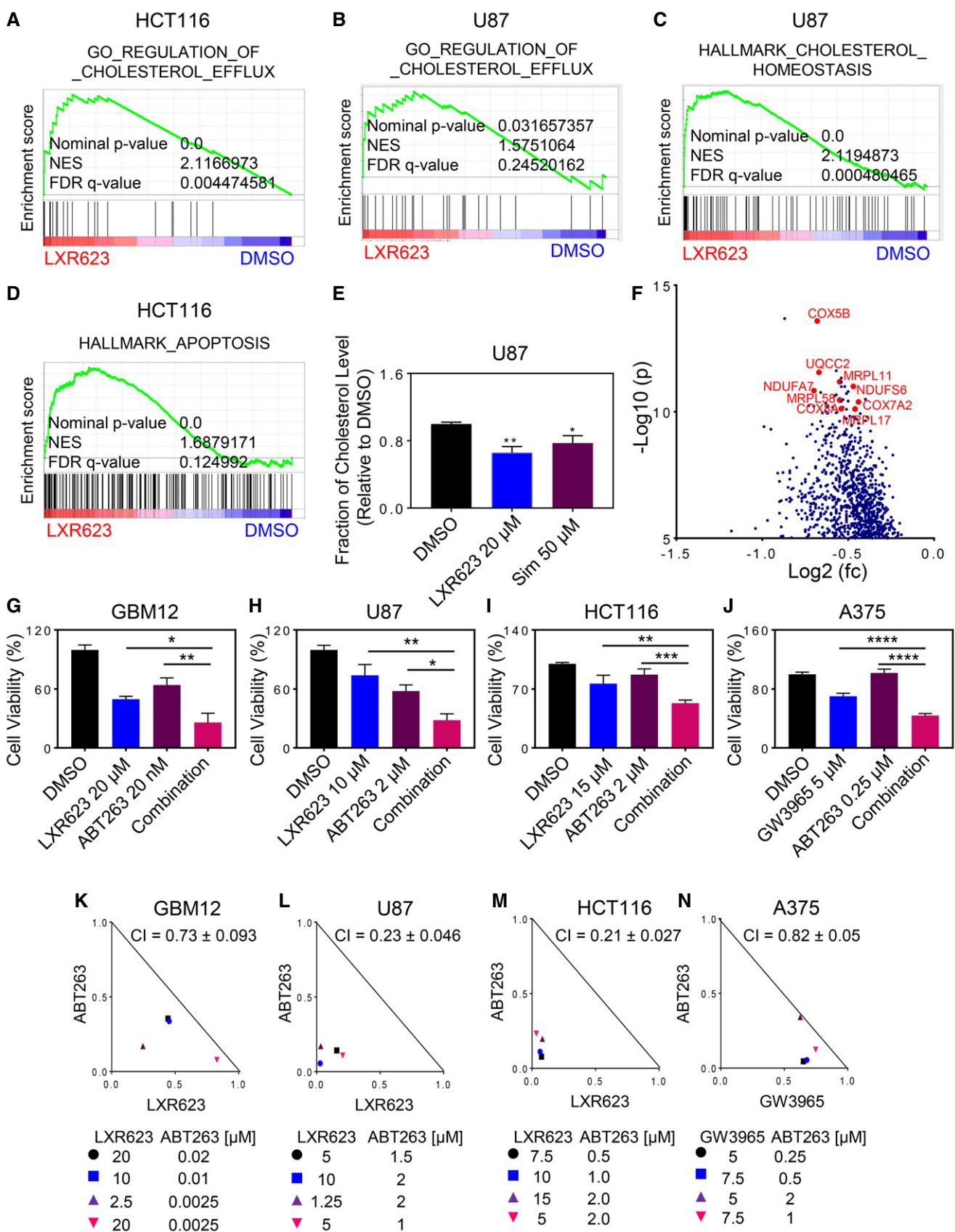

Figure 1.

**Figure 1. Combined treatment with ABT263 and LXR623 results in a synergistic anti-proliferative effect across solid tumor cells.**

A–D  HCT116 and U87 cells were treated with 20 μM LXR623 for 24 h. Transcriptome and gene set enrichment analysis was performed. Shown are enrichment plots. NES: normalized enrichment score.

E  U87 GBM cells were treated with 20 μM LXR623 or 50 μM simvastatin (Sim) for 48 h. Thereafter, lysates were collected and analyzed for total cholesterol levels. Shown are means and SD ($n = 3$). *$P = 0.0145$, **$P = 0.0019$. Statistical significance was determined by one-way ANOVA.

F  Shown is a volcano plot of patients from the TCGA (database) with high vs. low levels of ABCA1. A close-up of the most significantly down-regulated genes is presented (mRNAs related to the electron transport chain and mitochondrial metabolism). The x-axis highlights the fold changes ($\log_2$ scale) (down-regulated genes, negative values), whereas the y-axis indicates the level of statistical significance (shown in $-\log_{10}$ format). Figure EV1 contains the entire volcano plot. fc: fold changes; p: P-value.

G–J  GBM12 GBM cells (short-term patient-derived xenograft), U87 GBM cells, HCT116 colonic carcinoma cells, and A375 melanoma cells were treated with the indicated concentrations of ABT263, LXR623/GW3965, or the combination for 72 h. Thereafter, cellular viability was analyzed and statistical analysis was performed. Shown are means and SD ($n = 3$–4). GBM12: *$P = 0.0145$, **$P = 0.0019$; U87: *$P = 0.0108$, **$P = 0.0012$; HCT116: **$P = 0.0037$, ***$P = 0.0003$; A375: ****$P < 0.0001$. Statistical significance was determined by one-way ANOVA.

K–N  CI (Combination Index) value indicates whether the drug combination is additive (CI value = 1.0), synergistic (CI value < 1.0), or antagonistic (CI value > 1.0). Normalized isobolograms are presented for the combination treatments in the various solid tumor cell lines.

response to LXR agonist treatment. Our results show that LXR623 up-regulates the expression of the pro-apoptotic Bcl-2 family member, Noxa, in glioblastoma, colon cancer, and melanoma cell lines (Fig 3A and Appendix Fig S3B). Akin to LXR623, GW3965 caused an increase in Noxa protein levels as well (Appendix Fig S3A). These increases were associated with an up-regulation of the LXR downstream target, ABCA1, implying a potential relationship between cholesterol efflux and Noxa protein levels (Fig 3B).

**Specific knockdown of Noxa and BAK protects from ABT263- and LXR-mediated apoptosis**

To assess as to whether or not Noxa up-regulation is pivotal for the cell death induction of the combination therapy (ABT263 and LXR623), we silenced the expression of Noxa, using two Noxa-specific siRNAs, in LN229 GBM cells (Fig 3C). Knockdown of Noxa was confirmed by capillary electrophoresis and led to a significant attenuation of ABT263-/LXR263-mediated apoptosis (Fig 3C and D and Appendix Fig S3C). Since Noxa binds to Mcl-1 and facilitates BAK release, we silenced the expression of BAK as well. Suppression of BAK was confirmed by capillary electrophoresis (Fig 3D). As expected, silencing of BAK protected LN229 cells from killing by the combination treatment of ABT263 and LXR623 (Fig 3C and Appendix Fig S3C). These results establish Noxa and BAK as key elements for the combination treatment to exert apoptotic effects. To gain more insight into the molecular events, we conducted

immunoprecipitation experiments. We immunoprecipitated BAK in the four different conditions. In the control conditions, most BAK protein was bound to both Mcl-1 and Bcl-xL (Fig 3E). Upon treatment with LXR623, we found a reduction of BAK binding to Bcl-xL. As expected, this was even more pronounced in the ABT263 treatment condition in agreement with the major biological function of this BH3 mimetic. In both single treatment conditions, BAK was bound to Mcl-1. Notably, the combination treatment of ABT263 and LXR623 led to a significant displacement of BAK from both Bcl-xL and Mcl-1, in keeping with the synergistic cell death induction elicited by the combination treatment (Fig 3E).

**Activation of LXRβ decreases ATP levels accompanied by an integrated stress response**

To understand as to how LXR agonists lead to this substantial increase in Noxa levels, we reasoned that it is likely to be related to an induction of an integrated stress response, which can be induced by various means, including energy deprivation. Our transcriptional and gene set enrichment analysis pointed toward a state of energy deprivation, inhibition of mitochondrial metabolism, and an integrated stress response (Appendix Fig S4A and Fig EV1D–F). Therefore, we tested the hypothesis that LXR agonists reduce ATP levels and found that U87 GBM and HCT116 colon carcinoma cells treated with LXR623 or the related LXR agonist, GW3965, showed lower ATP levels in a concentration-dependent manner, while the cells

**Figure 2. Combined treatment with LXR623 and BH3 mimetics yields enhanced induction of apoptosis.**

A  U87, T98G, and LN229 cells were treated with 1 μM ABT263, 20 μM LXR623, or the combination of both for 72 h. MeWo melanoma and HCT116 colon carcinoma cells were treated with 1 μM ABT263, 10 μM LXR623, or the combination of both for 72 h (MeWo) and for 48 h (HCT116). Thereafter, cells were stained with annexin V/propidium iodide and analyzed by multi-parametric flow cytometry. Shown are means and SD ($n = 3$). ***$P = 0.0007$, ****$P < 0.0001$. Statistical significance was determined by one-way ANOVA.

B  Stem-like GBM cells, NCH644, NCH421k, and NCH690, were treated with 1 μM ABT263, 20 μM LXR623, or the combination of both for 48 h. Thereafter, cells were stained with annexin V/propidium iodide and analyzed by multi-parametric flow cytometry. Shown are means and SD ($n = 3$). ****$P < 0.0001$. Statistical significance was determined by one-way ANOVA.

C, D  U87 (72 h treatment) and HCT116 (48 h treatment) were treated with 1 μM ABT263, 20 μM (U87) or 10 μM (HCT116) LXR623, or the combination of both, stained with TMRE and analyzed by flow cytometric analysis for dissipation of mitochondrial membrane potential. Shown are means and SD ($n = 3$). ****$P < 0.0001$. Statistical significance was determined by one-way ANOVA.

E  HCT116 colonic carcinoma cells were treated with ABT263, LXR623, or the combination of both. Whole protein lysates were collected and subjected to capillary electrophoresis for the expression/cleavage of PARP, total caspase-9 (CP9), and Vinculin. CF: cleaved fragment; FL: full length.

F  HCT116 colonic carcinoma or NCH644 glioblastoma stem cells were treated with selective BH3 mimetics, WEHI-539 (Bcl-xL inhibitor), ABT199 (Bcl-2 inhibitor), or A1210477 (Mcl-1 inhibitor) in the presence or absence of LXR623 for 48 h. Thereafter, cells were labeled with annexin V/propidium iodide and analyzed by multi-parametric flow cytometry.

Source data are available online for this figure.

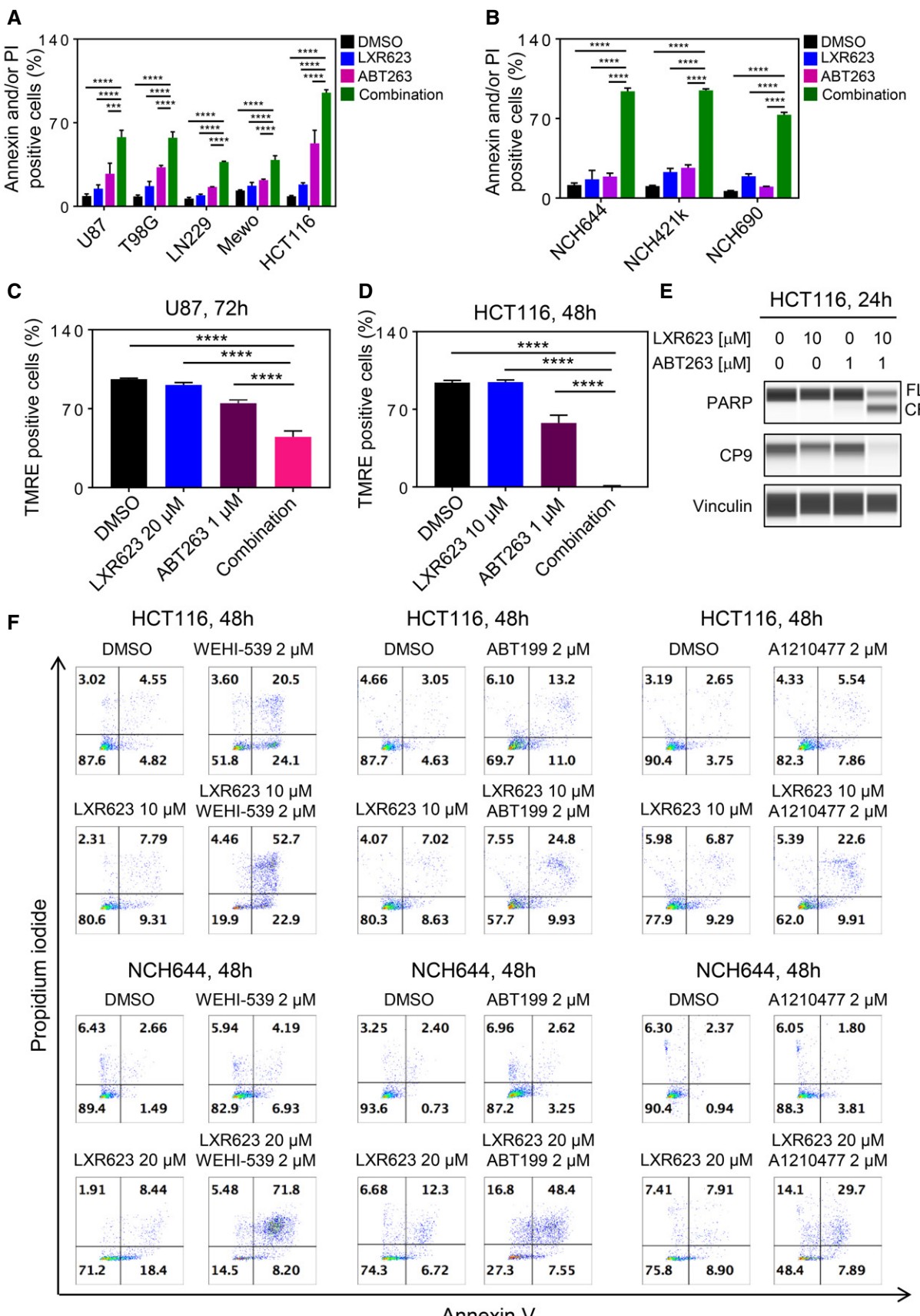

Figure 2.

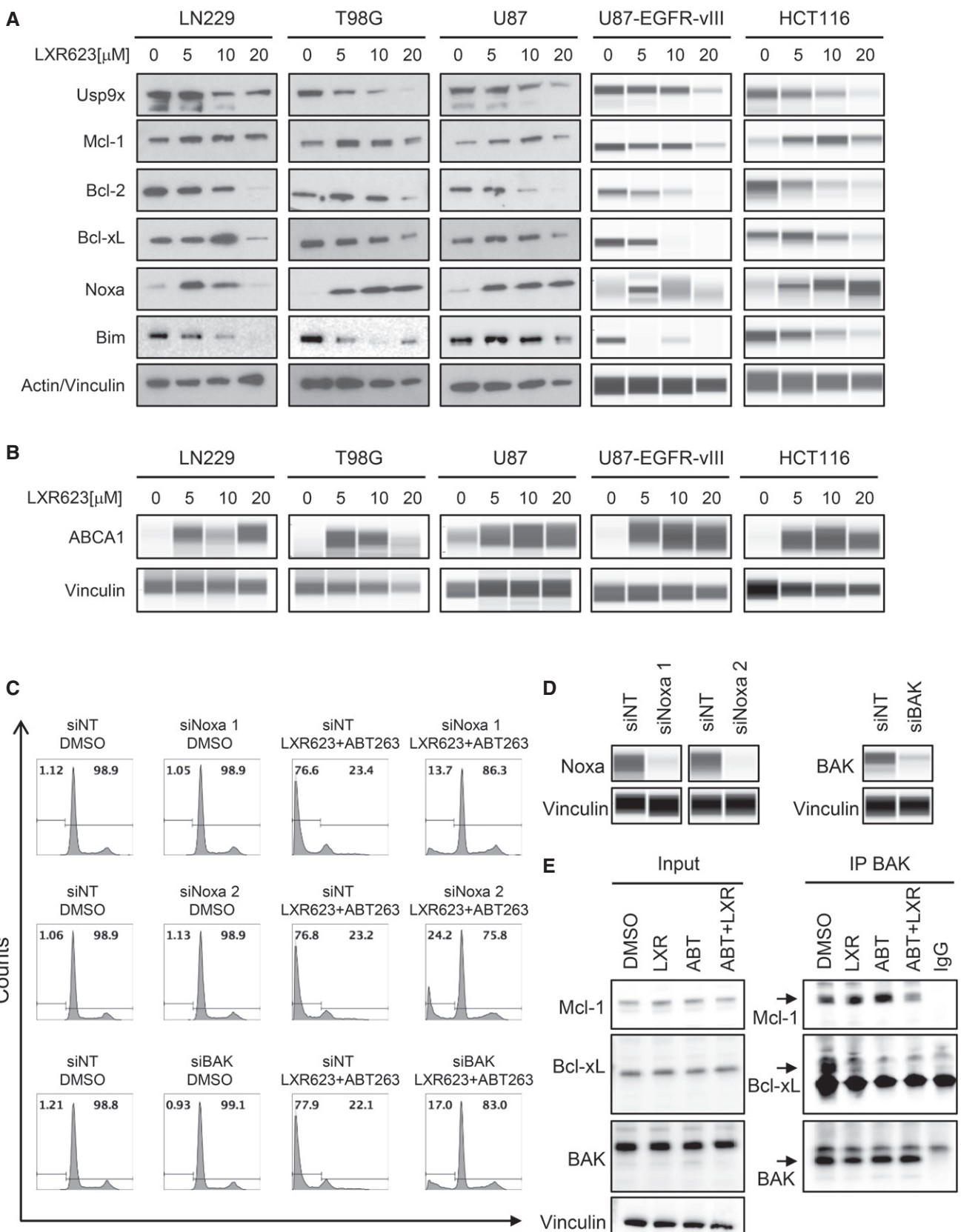

Figure 3.

**Figure 3.  Treatment with LXR623 regulates the expression of pro-apoptotic Noxa, and increased Noxa levels mediate enhanced sensitivity to BH3 mimetics.**

A    LN229, T98G, U87, U87-EGFRvIII, and HCT116 cells were treated with increasing concentrations of LXR623 for 72 h. Thereafter, protein lysates were collected and subjected to standard Western blot (LN229, T98G, and U87) or capillary electrophoresis (U87-EGFRvIII and HCT116) for the expression of Usp9X, Mcl-1, Bcl-2, Bcl-xL, Noxa, BIM, and/or actin and Vinculin. In U87-EGFRvIII and HCT116, Vinculin was used in lieu of actin.

B    LN229, T98G, U87, U87-EGFRvIII, and HCT116 cells were treated with LXR623 as described in (A). Thereafter, protein lysates were collected and subjected to capillary electrophoresis for the expression of ABCA1 and Vinculin.

C    LN229 were transfected with siNT, siNoxa 1, siNoxa 2, or BAK siRNA for 72 h. Thereafter, cells were treated with the combination treatment of 1 μM ABT263 and 20 μM LXR623 for another 24 h. After conclusion of the treatment, cells were harvested, fixed, stained with propidium iodide, and analyzed by flow cytometry for DNA fragmentation.

D    LN229 were transfected as described in (B). Thereafter, whole-cell protein lysates were collected and analyzed by capillary electrophoresis for the expression of Noxa, BAK, and Vinculin.

E    LN229 GBM cells were treated with 10 μM LXR623 (LXR), 1 μM ABT263, or the combination for 48 h. Thereafter, protein lysates were prepared and immunoprecipitated with an antibody against BAK. Standard Western blotting was performed (immunoprecipitation and the corresponding inputs) with antibody against BAK and Mcl-1. The arrows highlight the specific protein bands, while stars indicate the immunoglobulin light chains.

Source data are available online for this figure.

were completely viable (Fig 4A and B, and Appendix Fig S4B and E). In addition, we validated the reduction in ATP levels by a second methodology, involving LC/MS, showing similar findings (Fig 4C). Consequently, low ATP levels coupled with high AMP levels drive the phosphorylation of AMPK at threonine 172. To validate this claim, U87 cells were treated with increasing concentrations of LXR623. Thereafter, cells were analyzed by capillary electrophoresis for total AMPK and phosphorylated AMPK (threonine 172). As anticipated, LXR623 increased the amount of phosphorylated AMPK in a concentration-dependent manner (Fig 4D). Given that LXR agonists suppress ATP levels, we posed the question as to whether or not interference with cellular respiration suffices to sensitize cells to BH3 mimetic-mediated cell death. To this end, U87 and HCT116 cells were treated with oligomycin [complex V inhibitor (ATP synthase)], ABT263, or the combination in U87 for 48 h. Thereafter, cells were stained with annexin V/propidium iodide and analyzed by multi-parametric flow cytometry analysis. We observed a remarkable synergistic induction of apoptosis in cells treated with the combination treatment of oligomycin and ABT263 (Appendix Fig S5A and B), in keeping with our hypothesis that energy depletion (ATP loss) and/or inhibition of mitochondrial

metabolism renders solid tumor cells susceptible to Bcl-xL inhibition.

Since reduced energy levels may result in the induction of an integrated stress response (Kaufman *et al*, 2002; Chae *et al*, 2012), we tested this hypothesis by determining the levels of ATF4, GRP78, and related markers following treatment with either LXR623 or GW3965 and noted an increase in ATF4 at 7 and 24 h (Fig 4E, G, and I, and Appendix Fig S5D). ATF4 is a bona fide transcription factor and pivotal regulator that is tightly linked with stress response signaling that executes metabolic reprogramming, regulating cell growth and survival. For these implications, it appears likely that ATF4 may be a regulator of metabolic and cell death adaption following modulation of LXR signaling in cancer cells. We also interrogated our GSEA and found that LXR623 induces a transcriptional signature of ER stress (Appendix Fig S5C). To further corroborate our findings related to ER stress, we treated U87 and HCT116 cells with LXR agonists and isolated mRNA. Thereafter, we performed real-time PCR analysis of various ER-stress markers (ATF3, ATF4, XBP1, CEBPB, GRP78, and CHOP), demonstrating a consistent increase, in keeping with a profound activation of ER-stress signaling (Appendix Fig S5E–G). In agreement with the ER-stress response

**Figure 4.  Activation of liver-X-receptors cause energy deprivation and activate endoplasmic reticulum stress signaling, culminating in an increase in Noxa protein levels.**

A    U87 cells were treated with 20 μM LXR623 for 7 h and 24 h. Thereafter, cells were stained with annexin V/propidium iodide and analyzed by multi-parametric flow cytometry. The lower left quadrant indicates the percentage of viable cells.

B    U87 cells were treated with LXR623 as indicated for 7 h and analyzed for total ATP levels by a luminescence assay. To account for dead cells, assays were normalized to DNA content (cell number). Shown are means and SD ($n = 3$–4). ***$P = 0.0003$, ****$P < 0.0001$. Statistical significance was determined by two-sided Student's *t*-test.

C    U87 cells were treated with LXR623 as indicated for 24 h. Cell lysates were prepared and subjected to LC/MS for the determination of ATP levels. Shown are means and SD ($n = 3$). *$P = 0.0363$. Statistical significance was determined by two-sided Student's *t*-test.

D    U87 cells were treated with increasing concentrations of LXR623 for 7 and 24 h. Thereafter, whole-cell protein lysates were collected and analyzed by capillary electrophoresis for the expression of phosphorylated AMPK (threonine 172) and total AMPK.

E    U87 cells were treated with increasing concentrations of LXR623 in the presence or absence of exogenous ATP for 7 h. Thereafter, whole-cell protein lysates were collected and analyzed by capillary electrophoresis for the expression of ATF4.

F    U87 cells were treated with increasing concentrations of LXR623 for 24 h in the presence or absence of cholesterol. Thereafter, whole-cell protein lysates were collected and analyzed by capillary electrophoresis for the expression of phosphorylated AMPK (threonine 172) and total AMPK.

G, H  Cells were treated as in (F), and whole-cell protein lysates were collected and analyzed by capillary electrophoresis for the expression of ATF4 and Vinculin (G) and by the standard Western blot for the expression of Noxa and actin (H).

I    U87 cells were treated with increasing concentrations of LXR623. Thereafter, whole-cell protein lysates were prepared and analyzed for classical ER-stress markers, GRP78, ATF4, and Noxa by capillary electrophoresis.

J    U87 cells were transfected with non-targeting siRNA (n.t.-siRNA), ATF3-siRNA, ATF4 siRNA, or the combination of both ATF4 and ATF3 siRNAs. After 48 h, cells were treated with LXR623, and whole-cell protein lysates were harvested and analyzed by capillary electrophoresis for the expression of ATF3, ATF4, Noxa, and Vinculin.

Source data are available online for this figure.

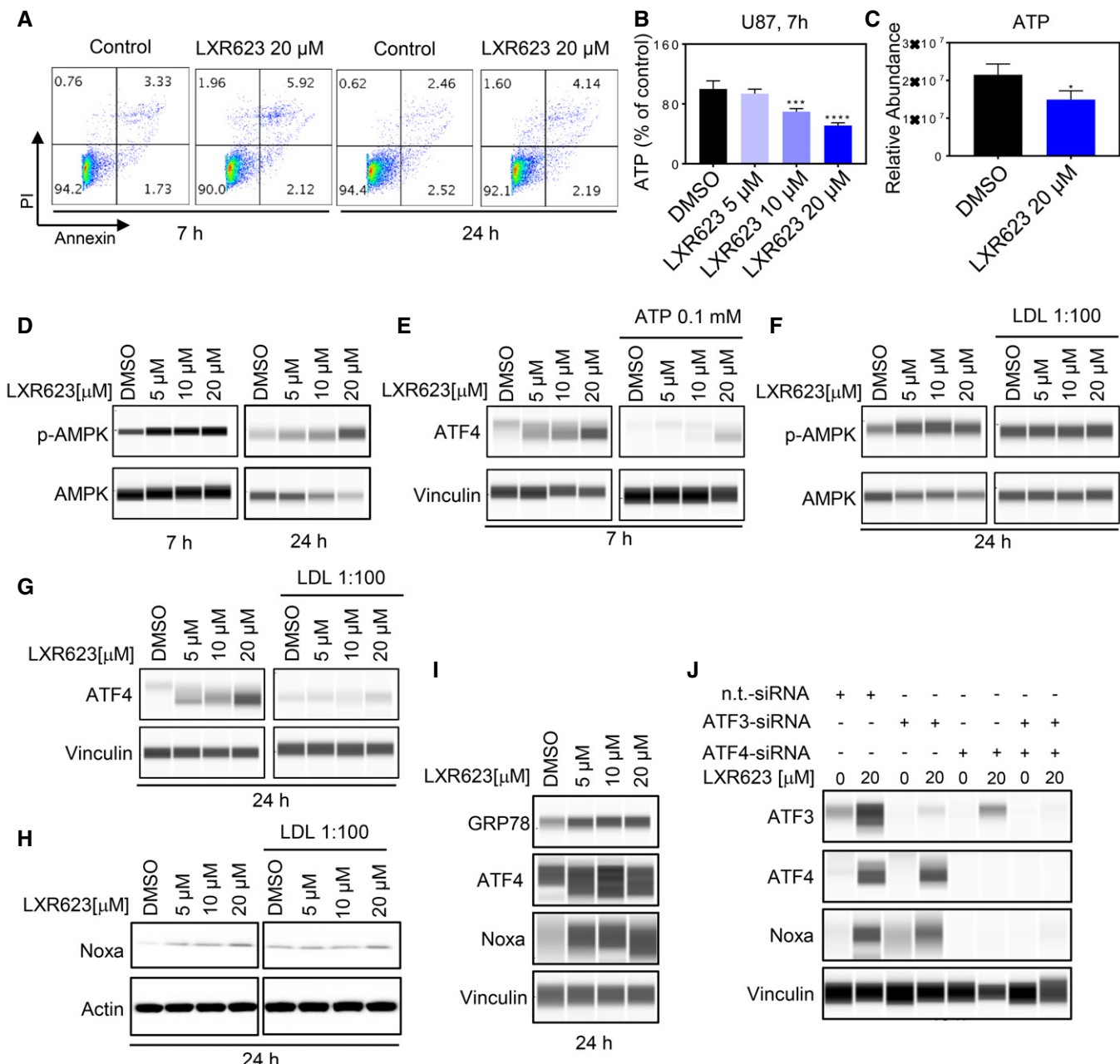

**Figure 4.**

transcriptional signature, we detected an increase in Noxa mRNA, suggesting that Noxa is in part regulated in a transcriptional manner upon LXR activation (LXR623 and GW3965). This increase was more settled in HCT116 cells treated with LXR623 (Appendix Fig S5F). In like manner, oligomycin recapitulated the effects of LXR activation with increases in phosphorylated AMPK (threonine 172), Noxa, and ER-stress-related markers, GRP78, ATF4, and ATF3 in U87 and HCT116 cells (Appendix Fig S5H and I).

To connect the underlying mechanisms of cholesterol and ATP depletion elicited by LXR activation, we treated U87 GBM cells with LXR623 in the presence or absence of exogenous ATP and observed that ATP partially rescues from LXR623-mediated increase in ATF4

protein levels (Fig 4E). Similar, albeit more pronounced findings were seen in the presence of cholesterol, which almost completely abrogated LXR623-driven ATF4 and Noxa increase and even reversed enhanced phosphorylation of AMPK (Fig 4F–H). These findings suggest a role of cholesterol in the mechanism elicited by LXR agonists and moreover suggest that energy depletion is involved in the activation of integrated stress response. Given that cholesterol rescues from all features elicited by the drugs (energy depletion, ATF4, and Noxa increase), it appears likely that cholesterol efflux is the initial event that is then followed by energy deprivation and activation of an ISR with subsequent up-regulation of Noxa.

ATF3 and ATF4 are well established as potent transcriptional inducers of Noxa (Wang *et al*, 2009; Karpel-Massler *et al*, 2017a). We noted that ATF4 levels preceded the increase in Noxa levels and 24 h following treatment with LXR623 ATF4 and Noxa levels correlated (Fig 4I and a detailed time course in Fig 6J). To connect our findings related to ISR and Noxa up-regulation, we targeted specifically the expression levels of ATF3 and ATF4 by siRNAs (both transcription factors have been described to regulate Noxa levels). To this end, U87, LN229 GBM, or HCT116 colonic carcinoma cells were transfected with ATF3, ATF4, or the combination of both siRNAs and thereafter exposed to LXR623 (Fig 4J, and Appendix Fig S5J and K). Thereafter, cells were treated with LXR623 for 48 h and analyzed for the expression of Noxa, ATF3, and ATF4. As previously seen, LXR623 increased Noxa protein levels in the presence of non-targeting siRNA. However, when ATF3 was silenced, we detected a partial attenuation of LXR623-driven Noxa increase (Fig 4J). Most prominently, knockdown of ATF4 suppressed LXR623-mediated Noxa increase, suggesting that ATF4 is the primary regulator of Noxa expression upon LXR receptor activation (Fig 4J, and Appendix Fig S5J and K). In keeping with these findings are our results related to the combined knockdown of ATF3 and ATF4, which was less potent to attenuate LXR623-driven Noxa up-regulation than ATF4 knockdown alone.

**LXRβ activation impairs glucose and glutamine metabolism**

Next, we determined the underlying mechanisms by which LXR activation reduces ATP levels in tumor cells. To this purpose, we performed a polar metabolite screen, followed by pathway analysis. This screen suggested that tricarboxylic acid (TCA) cycle as well as pyruvate metabolism is impaired in glioblastoma cells exposed to LXR623 (Fig 5A and B). Moreover, we detected a dysregulation of several metabolites of the TCA cycle (Fig 5C) and a reduction of NAD and NADH levels (Fig 5P).

Glucose and glutamine are the main energy sources in cultured tumor cells. Therefore, we conducted glucose and glutamine tracer analyses to appreciate the metabolic reprogramming in the context of LXR activation. We used uniformly labeled $^{13}C_6$-glucose to trace carbons into the TCA cycle and associated metabolites, such as aspartic acid and glutamate. We detected a decrease in labeling of TCA cycle intermediates by glucose carbons, including citric acid, oxoglutarate, succinic acid, fumaric acid, and malic acid. Under these conditions, citric acid is present in the form of several isotopologues, of which citric acid (m + 2) is indicative of glucose carbons derived from the pyruvate dehydrogenase reaction, which results in the loss of one carbon and suggests glucose oxidation. In contrast, citric acid (m + 3) originates from the pyruvate carboxylase reaction, which is one of the anaplerotic reactions to fuel the TCA cycle with carbons. LXR623 led to a decrease in citric acid (m + 2) and (m + 3) and to reduced labeling of oxoglutaric acid, succinic acid, fumaric acid, and malic acid (Fig 5D–H). In addition, we also found a decrease in labeling of non-essential amino acids, glutamic acid and aspartic acid, glutathione, and nucleotides (Fig 5I–N). Since aspartic acids react with carbamoyl phosphate as the initiating reaction for pyrimidine synthesis, it was not surprising to find a reduction in pyrimidine nucleotides (Fig 5L). In like manner, a reduction of purine nucleotides was detected as well given the participation of aspartate in purine synthesis (Fig 5M). The reduction in glutathione labeling by glucose carbons is explained by the fact that we found reduced glutamine labeling (by glucose carbons) and glutamine is part of the glutathione synthesis. A summary of the carbon tracing is provided in Fig 5O. Additional metabolites from the tracing analysis are presented in Appendix Fig S6A–E.

Given the reduction of labeling of the TCA cycle metabolites by glucose, we asked whether glutamine metabolism is altered as well since glutamine oxidation and anaplerosis have been described to be critical for cancer cells (Jiang *et al*, 2016). For these experiments, we used uniformly labeled $^{13}C_5$-glutamine. Glutamine enters the TCA cycle through conversion to glutamic acid and subsequently to α-ketoglutarate, which occurs via glutaminase and glutamate dehydrogenase and/or transaminases (Appendix Fig S6F). Once converted to α-ketoglutarate, it may either be oxidized or undergo

**Figure 5. Activation of LXR reprograms central carbon metabolism of tumor cells.**

A, B  U87 cells were treated with DMSO or LXR623 20 μM for 24 h. Thereafter, cells were processed for polar metabolite analysis by LC/MS. In panel (A), −log$_{10}$ *P*-values and log$_2$ fold change (fc) are shown. Highlighted in red are TCA cycle-related metabolites. In panel (B), metabolite pathway analysis was performed. Highlighted (red dots) are the tricarboxylic acid cycle and pyruvate metabolism as significant and impactful pathways altered by LXR623 treatment.

C  U87 cells were treated as in (A). Shown are the levels of TCA cycle metabolites. Shown are means and SD (*n* = 3).

D–H  U87 cells were incubated in DMEM (devoid of phenol red, glucose, pyruvate, and glutamine) supplemented with 25 mM U-$^{13}$C-glucose, 4 mM glutamine, and 1.5% dialyzed FBS in the presence or absence of 20 μM LXR623 for 24 h. Cells were then harvested for LC/MS analysis. The fractions of each different isotopologue of each metabolite were calculated (percentage of the entire pool). Shown are the isotopologues of the TCA cycle intermediates labeled by glucose carbons and non-labeled isotopologues (m + 0). Shown are means and SD (*n* = 3).

I, J  U87 cells were incubated in DMEM (devoid of phenol red, glucose, pyruvate, and glutamine) supplemented with 25 mM U-$^{13}$C-glucose, 4 mM glutamine, and 1.5% dialyzed FBS in the presence or absence of 20 μM LXR623 for 24 h. Cells were then harvested for LC/MS analysis. The fractions of each different isotopologue of each metabolite were calculated (percentage of the entire pool). Shown are the isotopologues of non-essential amino acids (glutamatic acid and aspartic acid). Shown are means and SD (*n* = 3).

K–N  U87 cells were incubated in DMEM (devoid of phenol red, glucose, pyruvate, and glutamine) supplemented with 25 mM U-$^{13}$C-glucose, 4 mM glutamine, and 1.5% dialyzed FBS in the presence or absence of 20 μM LXR623 for 24 h. Cells were then harvested for LC/MS analysis. The fractions of each different isotopologue of each metabolite were calculated (percentage of the entire pool). Shown are the isotopologues of glutathione and nucleotides. Shown are means and SD (*n* = 3).

O  Graphical depiction of glucose carbon tracing. Shown are the $^{13}$C-glucose carbons (red) and how they are transferred among molecules of the TCA cycle, amino acid biosynthesis, and purine/pyrimidine synthesis. Glucose is metabolized to pyruvic acid (m + 3) (three carbons labeled). When glucose is oxidized in the TCA cycle (m + 2), citric acid is produced (two carbons labeled). When glucose is used for anaplerosis, citric acid (m + 3) is produced (three carbons labeled). Glucose carbons are harnessed for the biosynthesis of glutathione either through the serine/glycine pathway or the TCA cycle via oxoglutaric acid and glutamate. The graphical presentation is representative for only one turn of the TCA cycle.

P  U87 cells were treated with DMSO or LXR623 20 μM for 24 h. Thereafter, cells were processed for polar metabolite analysis by LC/MS. Shown are the levels of NAD and NADH$_2$. Shown are means and SD (*n* = 4–6); statistical significance was determined by two-sided Student's *t*-test.

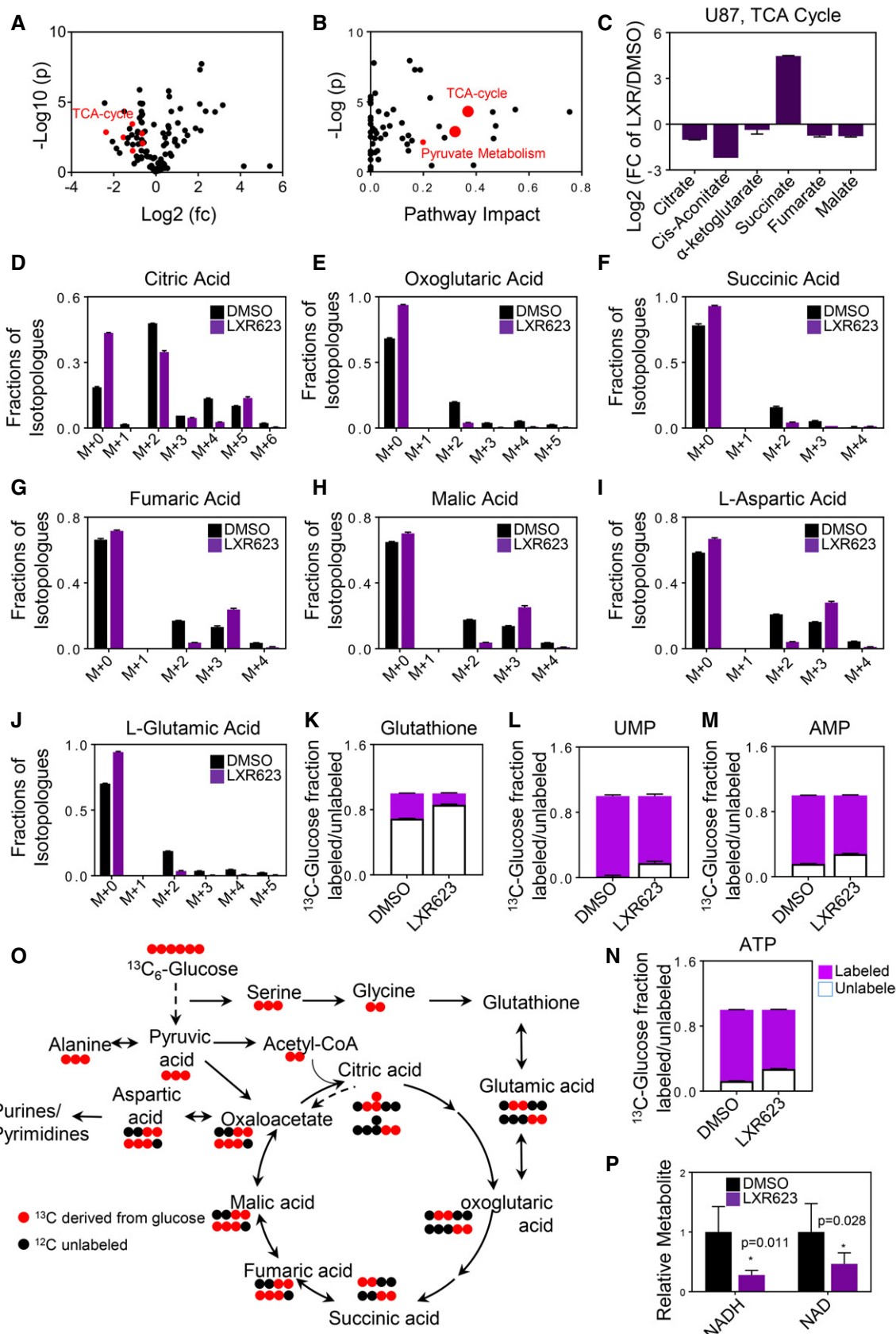

Figure 5.

reductive carboxylation, which means that α-ketoglutarate will be converted to citric acid (m + 5, isotopologue) in a reductive process. Consequently, almost the entire pool of glutamine is uniformly labeled in both control and LXR623 treatment groups (Appendix Fig S6G). LXR623 treatment led to a decrease in labeling of several TCA cycle metabolites by glutamine-derived carbons (Appendix Fig S6H–L). We noted a decrease in the oxidative metabolism of glutamine as indicated by reduced levels of citric acid (m + 4), fumarate (m + 4), and malate (m + 4) (Appendix Fig S6H–J). However, α-ketoglutarate (m + 5), succinate (m + 4), and glutamate (m + 5) labeling was increased (Appendix Fig S6H–M). In contrast, a relative enhancement of reductive carboxylation, involving increases of citrate (m + 5), fumarate (m + 3), malate (m + 3), succinate (m + 3), and amino acid aspartate (m + 3), was observed by LXR623 treatment (Appendix Fig S6H–N). While we noted a decrease in glutathione labeling from glucose, more glutamine carbons were incorporated into glutathione upon LXR623 treatment (Appendix Fig S6O). Due to the partial enhanced metabolism of glutamine (enhanced labeling of glutamate, α-ketoglutarate, and succinate), we tested whether LXR activation results in enhanced killing of glioblastoma cells under glutamine-deprived conditions. Our result suggests that glutamine deprivation sensitizes for LXR activation-driven cell death (Appendix Fig S6P). Considering the levels of essential and non-essential amino acids, we found that especially aspartate levels were suppressed, which is in keeping with earlier reports, demonstrating that under conditions of inhibition with OXPHOS, aspartate becomes critical for tumors to survive and grow (Appendix Fig S6Q; Molina *et al*, 2018). Overall, these findings are consistent with a profound reprogramming of the TCA cycle reactions by LXR agonists.

### The functionality of the electron transport chain is regulated by LXRβ in cancer but not in normal cells

The proper functionality of the TCA cycle relies on functional oxidative phosphorylation. Therefore, we conducted extracellular flux analysis and determined oxygen consumption rate (OCR) in multiple measurements (so-called mitochondrial stress assay), starting with basal OCR followed by acute injection of oligomycin (determination of OXPHOS-dependent ATP production or coupled respiration), the uncoupling reagent FCCP (determination of reserve and maximal respiratory capacity), and finally antimycin/rotenone (determination of mitochondria-related oxygen consumption). Stem-like GBM cells, NCH644, and established U87 and LN229 GBM cells treated with LXR623 displayed a significant reduction in basal OCR and OXPHOS-driven ATP production as compared to vehicle-treated control cells (Appendix Fig S7A–L). Additional experiments were conducted in HCT116 colon carcinoma cell lines, involving two LXR agonists, GW3965 and LXR623. LXR623 showed a dose-dependent reduction in basal OCR and coupled respiration (Appendix Fig S7M–O). To appreciate the timing of these findings, we conducted a time-course experiment. This experiment demonstrates that as early as 5 h LXR623 suppresses respiration and associated ATP production (coupled respiration; Fig 6A–F and Appendix Fig S7D–I). To account for the role of cholesterol in this phenomenon, we tested the impact of LXR623 on oxidative phosphorylation in the presence or absence of cholesterol and found that cholesterol fully reversed the effects by LXR623 on tumor cell respiration and associated parameters,

such as coupled respiration (ATP production by oxidative phosphorylation; Fig 6G–I). Overall, these findings indicate that LXR activation results in reprogramming of tumor cell metabolism with a reduction of OXPHOS.

To elucidate the role of LXR agonist-mediated suppression of oxidative phosphorylation on tumor cell survival, we undertook rescue experiments. To this purpose, we modulated the glucose concentration and found that as anticipated, low glucose concentrations significantly enhanced the killing effect of LXR623 (Appendix Fig S8A and B). In like manner, tumor cells cultured in galactose (which forces the cells to oxidize the galactose-derived glucose) displayed an increased susceptibility to LXR623 treatment (Appendix Fig S8C). Conversely, pyruvate and aspartate rescued from cell death mediated by LXR623, in keeping with earlier reports by others, showing that OXPHOS-deficient cells become auxotroph for pyruvate and dependent on the non-essential amino acid aspartate (Appendix Fig S8E and F). Finally, exogenous nucleotides were capable to provide a substantial rescue from LXR623-mediated cell death as well (Appendix Fig S8D). All in all, these observations favor that LXR agonist-driven inhibition of OXPHOS is at least partially relevant for their cytotoxic effects on tumor cells.

Next, we assessed the molecular underpinnings that likely lead to this fulminant reduction in OCR and derived measurements. To this end, we determined the protein levels of the respiratory complexes in a time-course experiment (Fig 6J). Our findings show that increasing concentrations of LXR agonists (LXR623 and GW3965) elicit a suppression of OXPHOS-related complexes as early as 5 h, coinciding with the suppression respiration (as detected by the extracellular flux analysis and accompanied by an increase in ATF4, followed by Noxa up-regulation; Fig 6J). Similar findings were made in HCT116 and stem-like GBM cells, NCH644 (Appendix Fig S8G–I). The prominent suppression of SDHB (at 24 h) also fits to our finding that glutamine oxidation is blocked at the level of succinate since in contrast to succinate, fumarate and malate display reduced levels of the respective (m + 4) isotopologues (Appendix Fig S6I, J, and L).

To determine the mechanism as to how a reduction of OXPHOS complexes occurs, we determined the stability of complex V in the presence or absence of LXR activation. We found that LXR623 destabilized complex V, suggesting a post-translational mechanism by which LXR activation suppresses respiratory complexes (Fig 6K). Due to the fact that cholesterol is important for the fluidity of membranes and for the proper functioning of membrane proteins, we evaluated the respiratory complexes in the presence or absence of LDL cholesterol and observed that LDL cholesterol protects from LXR623-mediated suppression of OXPHOS complexes (Fig 6L). Overall, these findings favor a direct regulation of oxidative phosphorylation by LXRs through down-regulation of the respiratory complexes and that likely enhanced cellular efflux of cholesterol destabilizes the respiratory complexes within the inner mitochondrial membrane.

In cancer cells, activation of LXRβ has been shown to be of highest relevance in the context of tumor growth and metastasis suppression (Villa *et al*, 2016; Wang *et al*, 2017). This holds also true for the model systems applied in this study. Therefore, we asked as to whether or not activation of LXRβ mediates suppression of respiration in tumor cells. To this purpose, we silenced the

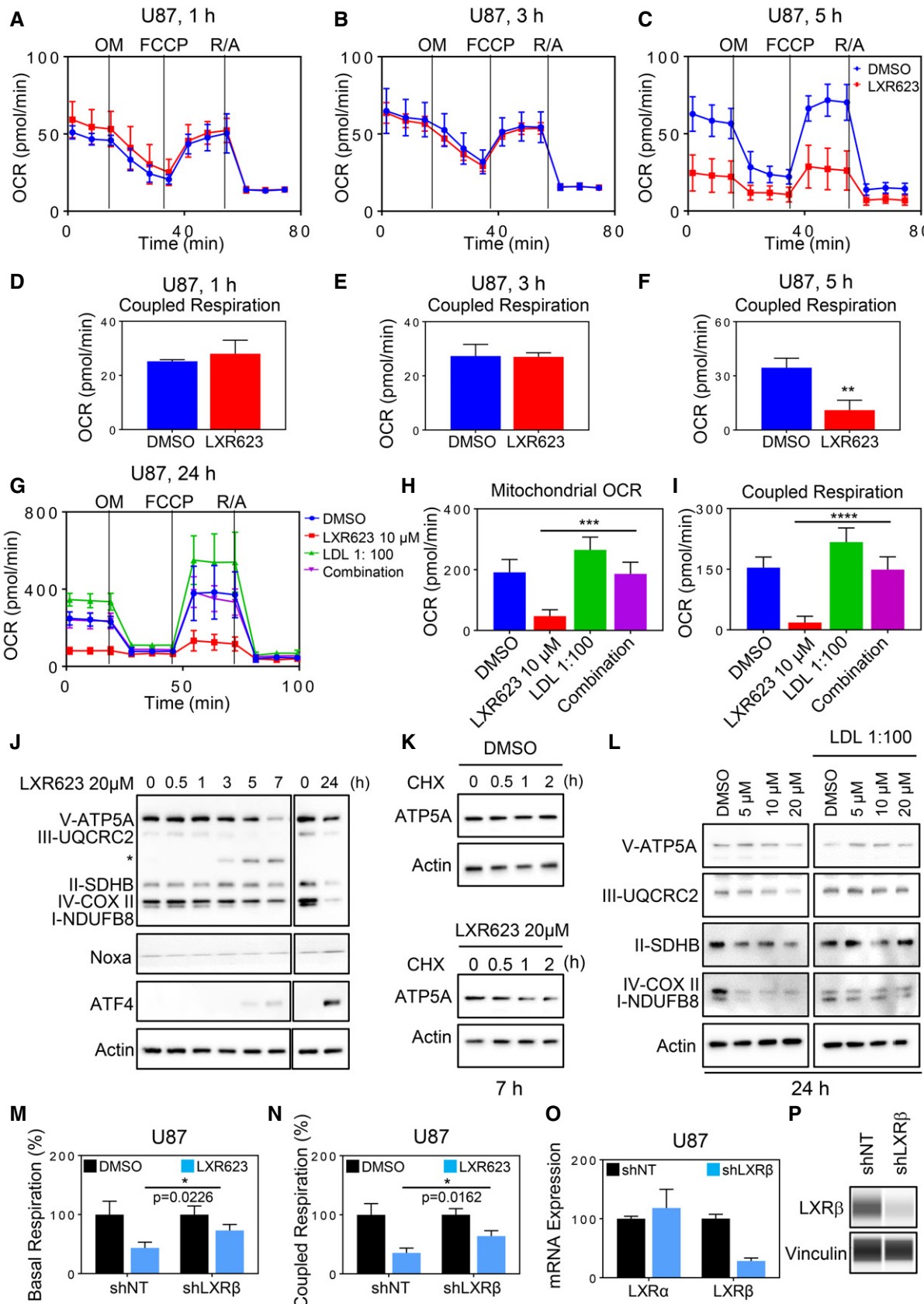

**Figure 6.**

**Figure 6.  Activation of liver-X-receptor mediates a state of energy deprivation through inhibition of oxidative phosphorylation (OXPHOS).**

A–F    U87 cells were treated with LXR623 for 1 h, 3 h, or 5 h and subjected to extracellular flux analysis on the Seahorse XFp instrument in the context of a
       mitochondrial stress assay (A–C). From this assay, coupled respiration was calculated (D–F). Shown are means and SD ($n$ = 3). **$P$ = 0.006. Statistical significance
       was determined by two-sided Student's $t$-test.

G–I    U87 cells were treated with LXR623, cholesterol, or the combination of both for 24 h and subjected to extracellular flux analysis on the Seahorse XFp instrument in
       the context of a mitochondrial stress assay (G). From this assay, mitochondrial OCR and coupled respiration were calculated (H, I). Shown are means and SD
       ($n$ = 5). ***$P$ = 0.0001, ****$P$ < 0.0001. Statistical significance was determined by one-way ANOVA.

J      U87 cells were treated with DMSO or LXR623 20 μM with the indicated time point. Thereafter, whole-cell protein lysates were collected and subjected to standard
       Western blot for the expression of the respiratory complexes (OXPHOS), Noxa, and ATF4. A star indicates an unspecific band.

K      U87 cells were treated with DMSO or LXR623 20 μM in the presence or absence of cycloheximide for 7 h. Thereafter, whole-cell protein lysates were collected and
       the protein levels of the components of complex V were analyzed by standard Western blot.

L      U87 cells were treated with increasing concentration of LXR623 in the presence or absence of cholesterol. Thereafter, whole-cell protein lysates were collected and
       analyzed the protein levels of the respiratory complexes (OXPHOS) by standard Western blot.

M–P    U87 cells transduced with non-targeting or LXRβ shRNA were treated with vehicle or 20 μM LXR623 for 24 h. Thereafter, extracellular flux analysis was performed
       in the context of a mitochondrial stress assay. From this assay, basal respiration and coupled respiration was calculated (M, N). mRNA and protein were collected
       and analyzed for the expression of LXRβ mRNA (O) or LXRβ protein by capillary electrophoresis (P). Shown are means and SD ($n$ = 3–4). Statistical significance was
       determined by two-sided Student's $t$-test.

Source data are available online for this figure.

expression of LXRβ (Fig 6O and P) and found that suppression of LXRβ levels attenuates the inhibitory effect of LXR623 on oxygen consumption rate and coupled respiration (Fig 6M and N), suggesting that LXRβ regulates OXPHOS in cancer cells. In addition, we over-expressed LXRβ and noted a suppression of oxidative phosphorylation, in keeping with our hypothesis that the regulation of LXR by itself modulates the activity of oxidative phosphorylation (Appendix Fig S9A and B). However, the effect of LXRβ over-expression was substantially less than the activation of the receptors by LXR agonists since only endogenous ligands (such as oxysterols) are available to activate the ectopically expressed receptors. Conversely, when the receptor was silenced we detected an increase in oxygen consumption rate (Appendix Fig S9C and D). Akin to the U87 cells, silencing of LXRβ suppressed LXR623-driven inhibition of oxygen consumption rate and ATP production in HCT116 cells, suggesting a primary role for LXRβ in these effects (Appendix Fig S9E–G).

In light of the substantial regulation of OXPHOS by LXR activation, we wondered as to whether or not non-neoplastic cells show a similar metabolic phenotype. To this purpose, human astrocytes were treated with LXR623 or GW3965 and analyzed for oxygen consumption rate. Remarkably, neither LXR623 nor GW3965 regulated the oxygen consumption rate, suggesting a tumor cell-specific phenomenon (Appendix Fig S9H and I).

Another major energetic pathway in cancer constitutes glycolysis. Therefore, it was tempting to determine the impact of LXR activation on glycolysis. Whereas we found a profound suppression of oxidative phosphorylation triggered by LXR activation, glycolysis was only marginally regulated by LXR activation, suggesting that LXR activation predominantly reprograms tumor cell metabolism by regulation of oxidative metabolism and that glycolysis is relatively elevated as compared to oxidative phosphorylation (Appendix Fig S9J and K).

## ABT263 and LXR mediate enhanced growth reduction of a colon cancer xenograft, a glioblastoma patient-derived xenograft, and a BRAF V600E-mutated melanoma xenograft in nude mice

To demonstrate that Bcl-xL inhibition and activation of LXR are a treatment strategy that is of potential translational relevance, we assessed its impact on growth of xenograft tumors in immunocompromised mice. To this purpose, we used the HCT116 xenograft model and after tumor establishment four treatment groups were formed, consisting of groups that are subjected to treatments with vehicle, ABT263, LXR623, or the combination treatment of both (Fig 7A and B, and Appendix Fig S10A). Our findings indicate that both LXR623 and ABT263 reduced the growth of tumors in host animals. However, the combination treatment was even more potent to dampen tumor growth as compared to each compound on its own (Fig 7A and B, and Appendix Fig S10A).

To have a better understanding about the histological findings and the impact of the different compounds on cellular density, proliferation, and cell death (necrosis and apoptosis), we stained representative paraffin-embedded tumors with H&E. While vehicle, ABT263-, or LXR623-treated tumors displayed a dense cellular architecture and an abundance of mitosis, the combination treatment resulted in a marked lower cellular density accompanied by significantly more cell death in the form of necrosis as well as apoptosis and a reduction in mitotic figures (Fig 7G, H, and J–L). To confirm the impact on mitochondrial metabolism, we performed immunohistochemistry on tumors from our xenograft studies. LXR agonist-related treatments mediate a suppression of SDHB protein levels *in vivo* (Fig 7I), mirroring the *in vitro* findings.

To extend the findings obtained in the HCT116 colon cancer xenograft model, we next tested the various treatments in the setting of a patient-derived xenograft model of human glioblastoma (GBM43). Akin to the findings in the colon cancer model, we found that the combination treatment markedly reduced the tumor sizes in host animals as compared to single or vehicle treatments (Fig 7C and D, and Appendix Fig S10B).

Despite the significant advancements in immunotherapy and BRAF inhibitor therapies, melanomas still represent a challenge in oncology since these treatment modalities will certainly extend life expectancy, but not provide a cure or at least a transformation to a chronic disease stage. To this purpose, we tested our therapy in a model system of BRAF V600E mutant melanoma, using the A375 xenograft model. Akin to the findings above, the combination treatment of ABT263 and GW3965 reduced tumor growth significantly stronger than each compound on its own (Fig 7E and F, and Appendix Fig S10C), suggesting that overall our treatment

concept of inhibition of Bcl-2 family members along with activation of LXR agonists is a viable approach for a range of solid tumors.

## Discussion

Novel treatments targeting tumor-specific therapeutic windows are warranted. In this study, we have shown that LXR agonists, which regulate cholesterol efflux and influx within a cell, elevate the dependency of solid tumor cells on anti-apoptotic Bcl-2 family members. LXR agonists represent a potentially efficacious approach to target recalcitrant malignancies, which was exemplified in melanoma and glioblastoma (Pencheva et al, 2014; Villa et al, 2016). Of note is the previous finding that in a preclinical model of melanoma (syngeneic mouse model), the combination treatment of GW3965 and an immune checkpoint inhibitor exerted a significantly stronger anti-proliferative effect than each therapeutic on its own (Pencheva et al, 2014). Moreover, GW3965 was also effective in BRAF inhibitor-resistant melanoma models (Pencheva et al, 2014). Since melanoma mostly kills patient through distant metastasis, it was an important finding that GW3965 suppressed migration of melanoma cells very efficiently in a LXRβ-dependent manner (Pencheva et al, 2014). Although both melanoma and glioblastoma cell cultures respond to GW3965 and LXR623, glioblastomas likely would benefit more from the usage of LXR623 over GW3965 since the former has been shown to cross the blood–brain barrier (Villa et al, 2016), which in part might be attributed to the fact that LXR623 has a smaller molecular weight as compared to GW3965. While healthy astrocytes are synthesizing cholesterol, GBM cells mostly rely on the uptake of cholesterol, which in turn renders glioblastomas potentially susceptible to LXR agonists (Villa et al, 2016).

While the efficacy of LXR agonists is quite remarkable, the mechanisms governing these effects are less well understood. Our study fills this gap at least partially by providing a mechanism and through that a novel efficacious combination therapy, involving GW3965 and LXR623. Because Bcl-2 family members are central

regulators of cell death in malignant tumors (Preuss et al, 2013; Souers et al, 2013; Chan et al, 2015; Faber et al, 2015; Hata et al, 2015; Johnson-Farley et al, 2015; Elgendy et al, 2016; Karpel-Massler et al, 2017c), we hypothesized that LXR agonists have an impact on their expression. While certain members of this family are more relevant in hematological malignancies, others maintain a pivotal role in solid tumors. Most prominently, in solid tumors Bcl-xL has been suggested to be of higher importance than Bcl-2. Our study is generally in line with this observation since we found that selective inhibition of Bcl-xL enhances the effects of LXR623 to a higher extent than inhibitors targeting Bcl-2. Our results reveal that LXR agonists potently elevate the expression of pro-apoptotic Noxa, thus shifting the expression ratio of Mcl-1 and Noxa. The literature unequivocally shows that Noxa interacts with Mcl-1 and antagonizes its function by promoting the release of BAK from Mcl-1 (Oda et al, 2000; Perez-Galan et al, 2007; Wang et al, 2009; Albershardt et al, 2011; Du et al, 2011; Premkumar et al, 2012; Qing et al, 2012; Yan et al, 2014). Mcl-1 levels are increased in solid malignancies and counteract apoptosis. In line with this overall scheme, we found that Noxa knockdown protects from cell death induced by the combination treatment of ABT263 and LXR623. These findings are supported by earlier observations, showing that high levels of Noxa sensitize for BH3 mimetics-mediated cell death. Noxa expression is regulated by multiple means, including endoplasmic reticulum stress and associated transcription factors, such as ATF3 and ATF4, which harbor an upstream open reading frame. We found that LXR activation drives ER stress with an increase in ATF3 and ATF4, which based on our rescue experiments and time courses is likely in part mediated by LXR623-/GW3965-mediated inhibition of ATP generation through interference with mitochondrial respiration.

Inhibition of oxidative phosphorylation by activation of LXRβ is an early, novel mechanistic observation that links this receptor to the regulation of oxidative energy metabolism. Rescue experiments suggest a significant role of oxidative phosphorylation on cell death induction by LXR agonists. Our study shows that activation of LXRβ results in global suppression of mitochondrial metabolism. These findings were supported by polar metabolite screening coupled with

**Figure 7. Combined treatment with ABT263 and LXR623 leads to a growth suppression of colon carcinoma xenograft tumors, glioblastoma patient-derived xenografts, as well as BRAF V600E-mutated melanoma xenografts.**

A, B $1 \times 10^6$ HCT116 colon carcinoma cells were implanted subcutaneously. Animals were treated intraperitoneally with vehicle, LXR623 (100–200 mg/kg), ABT263 (100 mg/kg), or both agents (3 days per week for 1.5 weeks). Tumor growth curves show the development of tumor size for each treatment group. Scatter plots display the quantitative representation of the tumor size among the different treatments toward the end of the experiment. Shown are means and SD ($n \geq 5$). *$P = 0.022$, **$P = 0.0063$, ****$P < 0.0001$. Statistical significance was determined by one-way ANOVA.

C, D GBM43 PDXs were implanted subcutaneously. Animals were treated intraperitoneally with vehicle, LXR623 (100 mg/kg), ABT263 (75 mg/kg), or both agents (3 days per week for 1.5 weeks). Tumor growth curves show the development of tumor size for each treatment group. Scatter plots display the quantitative representation of the tumor size among the different treatments toward the end of the experiment. Shown are means and SD ($n \geq 4$). *$P = 0.022$, **$P = 0.0031$. Statistical significance was determined by one-way ANOVA.

E, F A375 BRAF V600E-mutated melanomas were implanted subcutaneously. Animals were treated intraperitoneally with vehicle, LXR623 (100 mg/kg), ABT263 (75 mg/kg), or both agents (3 days per week for 1.5 weeks). Tumor growth curves show the development of tumor size for each treatment group. Scatter plots display the quantitative representation of the tumor size among the different treatments toward the end of the experiment. Shown are means and SD ($n \geq 5$). *$P = 0.0109$ (GW3965 vs. Combination), *$P = 0.0166$ (ABT263 vs. Combination), **$P = 0.0047$. Statistical significance was determined by one-way ANOVA.

G Related to the HCT116 colonic xenograft model, representative histopathological images (hematoxylin and eosin stain) are shown from each treatment group. Scale bar: 50 μm.

H, I Sections from the same treatment groups as in (G) were stained with TUNEL and SDHB, respectively. Displayed are representative images of the staining. Scale bar 50 μm.

J–L Sections from the experiment in (A) were quantified for apoptotic bodies, number of TUNEL-positive cells, and number of mitosis per multiple high-power fields. Shown are means and SD. In (J): *$P = 0.0298$, ***$P = 0.0004$. In (K): **$P = 0.0016$, ***$P = 0.0001$. In (L): **$P = 0.004$ (LXR623 vs. Combination), **$P = 0.0092$ (ABT263 vs. Combination). Statistical significance was determined by one-way ANOVA.

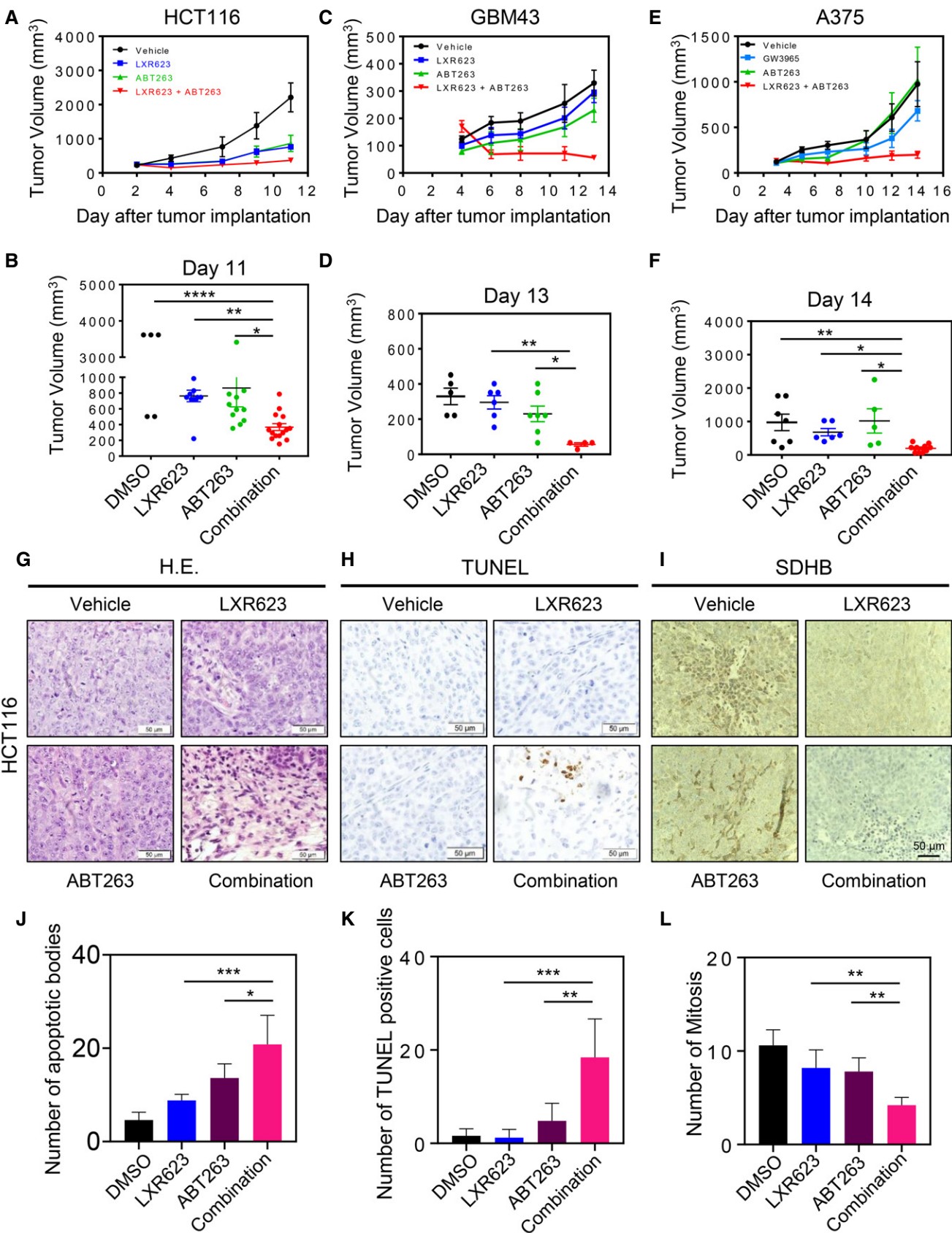

Figure 7.

metabolite set enrichment analysis and extracellular flux analysis that shows substantial inhibition of oxidative phosphorylation within 5 h of LXR agonist administration, which is fully rescued by exogenous cholesterol and partially by silencing of LXRβ. Western blot analysis confirms the suppression of respiratory complexes upon activation of LXRβ. In agreement, the TCA cycle metabolites were altered and carbon tracing experiments showed decreased labeling by glucose-derived carbons. We noted a decrease in labeling of the amino acid aspartate and nucleotides by glucose carbons, which is line with prior studies, that inhibition of OXPHOS depletes both aspartate and nucleotides, which in turn can elicit the induction of a potential DNA-stress response (Molina *et al*, 2018). Through our glutamine tracing experiment, we have found that LXR activation renders tumor cells partially more dependent on glutamine for their survival. Although we have not analyzed as to whether or not LXR leads to the emergence of DNA damage, we found a substantial decrease in NAD$^+$ levels, which could be explained through potential activation of the DNA-repair machinery, especially the PARP enzyme since PARP is known to require NAD$^+$ as a cofactor (Parrish *et al*, 2015). Remarkably, LXR activation does not suppress oxidative phosphorylation in human astrocytes, suggesting a tumor-selective phenomenon that might be harnessed therapeutically. To the best of our knowledge, these findings related to metabolism and LXR activation are novel and unique in tumor cells.

Finally, we demonstrate that LXR623 exerts anti-tumor activity *in vivo* in a colon xenograft model in nude mice without significant toxicity. This effect is enhanced further in the presence of the BH3 mimetic ABT263. These results confirm the *in vitro* findings and may raise hope that such a combination treatment might have implications for patients given the fact that both compounds have been tested in clinical trials. However, it should be noted that LXR623 displays side effects (Katz *et al*, 2009). Therefore, it may be possible that due to the combination treatment less drug will be necessary, thus reducing the toxicity. Additional toxicity studies are required for this point to be confirmed. Similar results were found in a patient-derived xenograft model system of glioblastoma and melanoma. It is tempting to speculate as to whether or not the selective Bcl-2 inhibitor, ABT199, would enhance the antiproliferative effects of LXR agonists as well. While this possibility seems less likely in solid tumors, it is highly conceivable in leukemias in which ABT199 is actively pursued. In addition, myeloid leukemia is known to be particular cholesterol-dependent. As discussed elsewhere, ABT199 is advantageous in that it does not affect platelets as compared to ABT263 because thrombocytes are Bcl-xL-dependent. Overall, our results suggest that LXR agonists, LXR623 and GW3965, affect the expression of Bcl-2 family members and thereby modulate the sensitivity of cancer cells to BH3 mimetics both *in vitro* and *in vivo*.

# Materials and Methods

### Reagents and antibodies

ABT263, WEHI-539, A1210477, GW3965, LXR623, ABT199, and adenosine triphosphate (ATP) were obtained from Selleckchem. Low-density lipoprotein (LDL) from human plasma was acquired from Thermo Fisher. L-Aspartic acid sodium salt monohydrate was purchased from Sigma. Reagents were dissolved in dimethylsulfoxide (DMSO) (10 mM stock solution) and stored at −20°C. Final concentrations of DMSO were below 0.1% ($v/v$). The following antibodies were used: ABCA1 (Abcam ab18180; 1:25), Mcl-1 (Cell Signaling Technology (CST) 5453SS; 1:500), BAK (CST 12105S; 1:500), Bcl-2 (Abcam ab59348; 1:500), BIM (CST 2933S; 1:500), Bcl-xL (CST 2764S; 1:500), Usp9X (CST 15751S; 1:1,000), Noxa (Millipore OP180; 1:500), β-actin (Sigma-Aldrich A1978, clone AC15; 1:2,000), ATF3 (CST 33593S; 1:500), ATF4 (CST 11815S; 1:500), Vinculin (Abcam ab129002; 1:200), AMPK (CST 5831S; 1:25), pAMPK (CST 2531S; 1:25), PAPRP (CST 9532S; 1:500), caspase-9 (CST 9502S; 1:500), GRP78 (CST 3177S; 1:25), HA (CST 3724S; 1:500), LXRβ (Abcam ab56237; 1:25), SDHB (Abcam ab14714, 1:25), OXPHOS (Abcam ab110411; 1:500), and secondary goat anti-mouse IgG-HRP-linked (SC2005) and secondary goat anti-rabbit IgG-HRP-linked antibodies (SC2004) were purchased from Santa Cruz Biotechnology, Inc.

### Maintenance of cells and transfections

The HCT116 (Horizon Discovery), U87 (ATCC), MDA-MB-231 (ATCC), MeWo (ATCC), LN229 (ATCC), and T98G (ATCC) cells were grown in DMEM supplemented with 10% FBS (Gemcell) and primocin (InvivoGen), and in a humidified air (5% CO$_2$) at 37°C. The cell line repositories confirmed the identities of the cell cultures and provided testing for mycoplasma contamination. Experiments were conducted in DMEM supplemented with 1.5% FBS and primocin (InvivoGen). The A375 cells were maintained in RPMI 1640 supplemented with 10% FBS (Gemcell) and primocin (InvivoGen). For the experiments, cells were cultured in RPMI medium 1640 containing 1.5% FBS (non-lipoprotein-deficient) and primocin (InvivoGen). In some experiments, lipoprotein-deficient serum (Sigma) was used. NCH644, NCH421k, and NCH690 glioma stem-like cells (Cell Line Service: CLS, Germany) were cultured in MG-43 medium (CLS, Germany) for both maintenance and experiments. Cells were transfected using either Lipofectamine RNAiMAX or Lipofectamine 3000 (Invitrogen) and harvested after 72 h after transfection. siPMAIP1 (siNoxa), siBAK, and siNR1H2 (siLXRβ) were purchased from Ambion. Non-targeting siRNA pool was purchased from Dharmacon. For the shRNA lentiviral particle transduction, cells were induced with Polybrene (Santa Cruz Biotechnology) and were selected with puromycin (Santa Cruz Biotechnology). shRNA NR1H2 (LXRβ) or non-targeting shRNAs were purchased from Santa Cruz Biotechnology. NR1H2 lentivirus was purchased from Applied Biological Materials Inc.

### Determination of intracellular cholesterol levels

100,000 cells (per well in a six-well plate) were treated with vehicle or the indicated drug compounds in DMEM supplemented with 1.5% lipoprotein-deficient serum for 48 h. Following drug exposure, cells were washed in PBS and collected. Following centrifugation, the pelleted cells were resuspended in 100 μl assay buffer (0.1 M potassium phosphate, 0.05 M NaCl, 5 mM cholic acid, 0.1% Triton X-100, pH 7.4) obtained from Thermo Fisher. Thereafter, cholesterol levels were measured by utilization of the Amplex Red Cholesterol Assay Kit (Thermo Fisher) in alignment with the standard protocol followed by plate reader analysis.

## Cell viability/proliferation assays

The indicated cell cultures were seeded in 96-well plates overnight before the treatment to permit proper attachment. On the next day, the treatments were carried out with vehicle or the indicated single drug compounds or combination treatments in DMEM supplemented with 1.5% FBS. Seventy-two hours after treatment, viability analysis was performed by using the "CyQUANT Cell Proliferation Assay®" following the standard protocols for adherent cells (provided by the manufacturer), followed by plate reader analysis. Cholesterol rescue experiments were carried out in 1.5% lipoprotein-deficient serum. The drug combination synergy analysis was calculated by the approach described by Chou–Talalay. Derived from median-effect equation (single drug dose–response related), it enables the computation of normalized isobolograms (Chou, 2010).

## Detection of mitochondrial membrane potential and apoptosis induction

Treatments were carried out in 12-well plates. Following treatments, cells were detached, washed, and labeled with annexin V and propidium iodide (BD Bioscience kit, #556547), and analyzed by flow cytometry. In other experiments, cells were detached, washed, and fixed. After fixation, cells were stained with propidium iodide (Propidium Iodide (PI)/RNase Staining Solution #4087, CST) and analyzed by flow cytometry. Labeling of cells with TMRE (tetramethylrhodamine ethyl ester perchlorate) was performed following treatments of cells in strict accordance with the protocol provided by the manufacturer (#13296, CST, Inc), and loss of mitochondrial membrane potential was recorded by flow cytometry. For the flow analysis, measurements were carried out on the LSR II flow cytometer instrument (BD, NJ, USA). The FlowJo software was used for generating graphs and data analysis (version 8.7.1, Tree Star, Ashland, OR).

## Determination of ATP levels

ATP levels were measured through LC/MS. The obtained values were normalized to protein levels. Alternatively, an ATP luminescence assay (CellTiter-Glo) was used and the values were normalized to DNA content to account for dead cells. The DNA content was measured by the CyQUANT Cell Proliferation Assay as described by manual of instruction from the manufacturer. The luminescence assays were conducted within a time frame of 7 h (with minimal loss of viability; Karpel-Massler et al, 2017b; Tang et al, 2017).

## Capillary electrophoresis and standard Western blotting

Following treatment, cells were harvested in Laemmli sample buffer (#1610737) or RIPA lysis buffer both supplemented with phosphatase and protease inhibitor cocktails (Thermo Fisher Scientific). Electrophoresis was performed on gradient gels (4–12% SDS–PAGE gels; Invitrogen). Wet transfers were carried out to PVDF membranes, followed by blocking in milk or BSA TBST (0.1% Tween 20) and incubation with the indicated primary antibodies overnight. Following washing, horseradish peroxidase-conjugated secondary antibodies

were used. Vinculin and β-actin served as loading controls and were used for normalization. Western blot emitted chemiluminescence was recorded on the Azure (C300) Imaging System (Azure Biosystems). The Wes instrument was utilized for capillary electrophoresis, including protein detection, in alignment with the protocols provided by the manufacturer (ProteinSimple).

## Reverse transcription real-time PCR analysis

The extraction of RNA was performed with the miRNAeasy Mini Kit (Qiagen). Reverse transcription of RNA was carried out using the "cDNA Synthesis kit" (OriGene). The PCR was performed with a master mix, containing SYBR green (Applied Biosystems). The real-time PCR machine (Quantabio) was programmed as follows: 95°C for 10 min, 40 cycles of 95°C for 15 s, 60°C for 30 s, and 72°C for 30 s. Upon completion of the run, data analysis was performed (delta–delta $C_t$ method). Genes of interest were normalized to 18S. Fold changes relative to the controls are provided. Forward and reverse primers are given in Table 1.

## Transcriptome analysis

Microarray and subsequent gene set enrichment analysis were performed as described earlier (Karpel-Massler et al, 2017b). The experiment used in this study was deposited at GEO: GSE110151 and GSE110152.

## Extracellular flux analysis

Oxygen consumption rate (OCR) and extracellular acidification rate (ECAR) were measured on the Seahorse XFp or XFe24 instrument, respectively. The mitochondrial stress test was performed, consisting of baseline measurements of OCR and in response to a defined set of drugs (oligomycin, FCCP, antimycin/rotenone), modulating oxidative phosphorylation. Alternatively, the glycolytic stress test was performed according to the manufacturer's instructions. Cells were seeded in XFe24 cell culture microplates (Agilent) at $3 \times 10^4$ cells/well in 500 μl of DMEM containing 5 mM glucose, 1 mM glutamine, 1 mM pyruvate, and 10% FBS and allowed to attach overnight. Cells were treated with reagents or corresponding solvents in the medium containing 5 mM glucose, 1 mM glutamine, 1 mM pyruvate, and 1.5% FBS in the following

**Table 1. Primers for real-time PCR.**

| Gene | Forward sequence | Reverse sequence |
|---|---|---|
| ATF3 | CGCTGGAATCAGTCACTGTCAG | CTTGTTTCGGCACTTTGCAGCTG |
| ATF4 | TTCTCCAGCGACAAGGCTAAGG | CTCCAACATCCAATCTGTCCCG |
| XBP1 | CTGCCAGAGATCGAAAGAAGGC | CTCCTGGTTCTCAACTACAAGGC |
| CEBPB | AGAAGACCGTGGACAAGCACAG | CTCCAGGACCTTGTGCTGCGT |
| GRP78 (HSPA5) | CTGTCCAGGCTGGTGTGCTCT | CTTGGTAGGCACCACTGTGTTC |
| CHOP (DDIT3) | GGTATGAGGACCTGCAAGAGGT | CTTGTGACCTCTGCTGGTTCTG |
| Noxa (PMAIP1) | CTGGAAGTCGAGTGTGCTACTC | TGAAGGAGTCCCCTCATGCAAG |

day. The assays were performed according to the manufacturer's instructions (Agilent).

## Liquid chromatography and mass spectrometry (LC/MS) analysis

U87 GBM cells were washed with 0.9% NaCl and were extracted with sequential addition of ice-cold methanol/water/chloroform (600 μl/300 μl/400 μl). Samples were vortexed and centrifuged. The lower lipid-containing layer and the upper aqueous layer were transferred to different tubes and evaporated to dryness under nitrogen. Dried extracts were stored at −80°C until analysis. Dried polar extracts were reconstituted in 100 μl water, and lipid extracts were reconstituted in 50 μl, 65:30:5 acetonitrile:2-propanol:water ($v/v/v$). Both polar and lipid samples were analyzed using LC-HRMS by the Whitehead Institute Metabolite Profiling Core Facility (Cambridge, MA) as previously described (Franco *et al*, 2016; Gui *et al*, 2016; Keckesova *et al*, 2017). Polar metabolites were analyzed by metabolite set enrichment analysis (http://www.metaboanalyst.ca).

## Metabolomics and isotope tracing

Cells were serum-starved in DMEM without glucose, glutamine, and phenol red (Thermo Fisher) for at least 1 h and then were incubated with either 25 mM D-glucose (U-$^{13}$C-glucose) (Cambridge Isotope Laboratories, Inc) or 4 mM L-glutamine (U-$^{13}$C-glutamine) for 24 h in the presence of 1.5% dialyzed FBS (Thermo Fisher). Cells were extracted in 80% methanol, and the supernatant was dried in a SpeedVac concentrator (Thermo Fisher). For metabolite analysis, the Exactive Orbitrap Mass Spectrometer (Thermo Scientific) coupled with a Vanquish UPLC system (Thermo Scientific) was used. The peak area for each metabolite was identified and quantitatively evaluated. Data analysis was facilitated by an in-house script designed by the Weill Cornell Medicine Core Facility (New York, NY) (Goncalves *et al*, 2018).

## Animal maintenance and subcutaneous xenograft models

For the *in vivo* experiments, 6- to 8-week-old female CrTac:NCr-*Foxn1$^{nu}$* were purchased from Taconic Biosciences. The mice were kept under controlled temperature in a 12-h (light)/12-h (dark) cycle, provided with water and food *ad libitum*. The animals were maintained in small groups (3–5 animals per cage). Following acclimation, the animals were subjected to the indicated experiments. $1 \times 10^6$ HCT116 colon carcinoma cells or A375 BRAF V600E-mutated melanoma cells suspended 1:1 in Matrigel® matrix (Corning Inc.) or GBM43 cryomush or cells (Sarkaria *et al*, 2006; Gupta *et al*, 2015; Parrish *et al*, 2015) were implanted subcutaneously into the flanks of 6- to 8-week-old CrTac:NCr-*Foxn1$^{nu}$* mice as described before (Karpel-Massler *et al*, 2017a). Tumors were measured with a caliper and sizes calculated according to the standard formula: (length * width$^2$) * 0.5. Treatment was performed intraperitoneally three times a week for 3 weeks. For intraperitoneal application, ABT263 and LXR agonists were dissolved in 10% DMSO, 32% Cremophor EL (Sigma), 8% ethanol (Pharmco-Aaper), and 50% PBS. Representative tumors were harvested, fixed in formalin, and thereafter embedded in paraffin. Thereafter, sections were cut that were stained for hematoxylin and eosin (H&E stain) by standard laboratory procedures.

### The paper explained

#### Problem

Advanced-stage solid cancers, such as colon carcinoma, melanoma, and glioblastoma, require efficient treatment approaches due to the short overall survival of patients with such tumors. Novel drug combination therapies may offer more efficient and selective targeting along with reduced toxicity than standard chemotherapies.

#### Results

Here, we show that inhibition of anti-apoptotic Bcl-2 family members together with activation of LXRs causes synthetic lethality in a variety of solid tumors, including colorectal carcinoma, melanoma, and glioblastoma. In xenografts, the treatment approach combining LXR623 and ABT263 reduces tumor growth more efficiently than single treatments. Using metabolomic approaches, we reveal a previously undescribed link between LXRβ receptor activation and regulation of the electron transport chain, leading to stasis of central carbon metabolism (tricarboxylic acid cycle), and culminating in energy loss and induction of an integrated stress response with activation of pro-apoptotic (cell death facilitating) Bcl-2-family member, Noxa.

#### Impact

Given that all pharmaceutical agents used in the current study have been clinically validated, our proposed treatment approach may be useful for patients suffering from malignant solid tumors.

## TCGA analysis

The TCGA analysis was performed by harnessing online computational tools, such as cBioPortal (Gao *et al*, 2013). The TCGA data for colonic adenocarcinoma and glioblastoma were taken into consideration and interrogated for high vs. low expression of ABCA1. Fold changes were derived, and statistical analysis was performed.

## Statistical analysis

Statistical significance was assessed by two-tailed Student's *t*-test or ANOVA (for multiple comparisons) using Prism version 5.04 (GraphPad, La Jolla, CA). A $P \leq 0.05$ was considered statistically significant. *$P < 0.05$; **$P < 0.01$; ***/****$P < 0.001$. The CompuSyn software (ComboSyn, Inc., Paramus, NJ) was used for the drug combination analysis including the calculation of the combination index (CI). CI (combination index) value indicates [as to] whether the drug combination is [either] additive (CI value = 1.0), synergistic (CI value < 1.0), or antagonistic (CI value > 1.0).

## Study approval

All procedures were in accordance with Animal Welfare Regulations and approved by the Institutional Animal Care and Use Committee at the Columbia University Medical Center.

# Data availability

The microarray used in this study was deposited at GEO: GSE110151 (https://www.ncbi.nlm.nih.gov/geo/query/acc.cgi?acc = GSE110151)

and GSE110152 (https://www.ncbi.nlm.nih.gov/geo/query/acc.cgi?acc = GSE110152).

Expanded View for this article is available online.

## Acknowledgements

This work was supported by the NIH NINDS K08NS083732, R01NS095848, R01NS102366, the 2017 American Brain Tumor Association Discovery Grant (DG1700013), and Louis V. Gerstner, Jr. Scholars Program (2017–2020). Trang Nguyen: American Brain Tumor Association Basic Research Fellowship (BRF1900018). Transcriptome analysis was supported by the CTSA grant UL1-TR001430 to the Boston University Microarray and Sequencing Resource Core Facility. These studies used the resources of the Cancer Center Flow Core Facility funded in part through Center Grant P30CA013696.

## Author contribution

TTTN, CTI, PC, and MDS designed research; TTTN, CTI, ES, CS, CT, YZ, and EB conducted the experiments. TTTN, CTI, ES, and MDS analyzed the data. TTTN, CTI, ES, CS, M-AW, GK-M, PC, and MDS helped with writing, review, and/or revision of the article. MJS-Q, GK, and CMQ provided administrative, technical, or material support (i.e., reporting or organizing data, constructing databases). MDS supervised the study.

## Conflict of interest

The authors declare that they have no conflict of interest.

## For more information

American Brain Tumor Association (ABTA): https://www.abta.org/
cBioPortal: https://www.cbioportal.org/

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
