## [Review Process File · EMBO Molecular Medicine]

Activation of LXR β Inhibits Tumor Respiration and is Synthetically Lethal with Bcl-xL Inhibition

Trang Thi Thu Nguyen, Chiaki Tsuge Ishida, Enyuan Shang, Chang Shu, Consuelo Torrini, Yiru Zhang, Elena Bianchetti, Maria J. Sanchez-Quintero, Giulio Kleiner, Catarina M. Quinzii, Mike-Andrew Westhoff, Georg Karpel-Massler, Peter Canoll and Markus D. Siegelin.

Review timeline:

Submission date:	17 th April 2019
Editorial Decision:	17 th May 2019
Author Correspondence	28 th May 2019
Editorial Correspondence	4 th June 2019
Revision received:	24 th June 2019
Editorial Decision:	30 th July 2019
Revision received:	1 st August 2019
Accept:	6 th August 2019

Editor: Lise Roth

Transaction Report:

1st Editorial Decision

17th May 2019

Thank you for the submission of your manuscript to EMBO Molecular Medicine. We have now received feedback from two of the three reviewers who agreed to evaluate your manuscript. Given that referee 1 is not responsive despite several chasers, and that both referees 2 and 3 are overall positive, we prefer to make a decision now in order to avoid further delay in the process. If referee 1 returns his/her comments within the next 2 weeks, we will forward these comments to you, but we will not ask for any further reaching experiment.

As you will see from the reports below, both referees acknowledge the interest of the study. However, they also raise substantial concerns on your work, which should be convincingly addressed in a major revision of the present manuscript. Particular attention should be given to increasing the clinical relevance of the study, by adding clinical data and/or data from a transgenic mouse model.

Addressing the reviewers concerns in full will be necessary for further considering the manuscript in our journal, and acceptance of the manuscript will entail a second round of review.

REFeree REPORTS

Referee #2 (Comments on Novelty/Model System for Author):

This study is of interest for medical questions surrounding cancer treatment. However, given that biological modeling is restricted to cancer cell lines, the clinical relevance is limited.

Referee #2 (Remarks for Author):

Nguyen T. et al. report inhibitory responses to LXR β activation and BH-3 mimetics on growth of

various solid tumor derived cell lines including colon carcinoma, melanoma and glioblastoma. The authors conducted a molecular investigation to decipher how LXR ligands leads to downregulate OXPHOS pathway in tumor cells. They provide strong evidence that these effects are sustained by the upregulation of NOXA in an ATF4-dependent fashion and impact on the energy balance. In consequence TCA cycle is strongly downregulated that sensitive cells to BH3-mimetics. Technical approaches are relevant and support conclusions drawn. Research is well conducted. Overall, proposed article provides relevant evidences that sustain the synergistic effects of both targeting LXR signaling and Bcl-xL family in order to induce apoptosis of tumor cells. The focus on OXPHOS pathway in cancer show a growing interest in the field. Cancer cell sensitivity recurrence to chemotherapeutic agent using mitochondrial processes is a hot topic. Thus, this article represents a significant contribution.

The main weakness of this study come from that biological modeling is restricted to cell line using. Article will gain impact with transgenic mouse model (APCmin for colon cancer for example), as well as clinical data that could support GSEA results obtained on cell culture experiments. Especially, could we observe a correlation between the cluster of LXR target genes and mitochondrial-related gene expression described in Suppl Figure 1C-E in patient datasets.

Minor comments:

- *Figures and Suppl Figures numeration is heterogeneous. Figure call needs to be in the correct order. Figure 1A 1C and 1B 1D needs to be Figure 1A 1B and 1C 1D. Similarly, panel number needs to be update for Suppl Figure 1, 5, 6 and 7 regarding the manuscript text apparition.
- *The section untitled "Activation of LXR β induces energy deprivation by shutting..." is too long, more than 8 pages! This part of the manuscript needs to be shortened as well as separated in various sections with only one message for each of them.
- * Given that authors claimed "These results nicely reinforce our in vitro results and raise hope that such a combination treatment might have implications for patients given the fact that both compounds have been tested in clinical trials", the following reference missing: Katz A, et al. Safety, pharmacokinetics, and pharmacodynamics of single doses of LXR-623, a novel liver X-receptor agonist, in healthy participants. *J Clin Pharmacol.* 2009 Jun;49(6):643-9. doi: 10.1177/0091270009335768. Authors should soften this comment regarding LXR623 exhibit significant side effects.
- *The use of 20 μ M LXR623 is questionable as we could expected to be out of the range of LXR specific effects.

Major comments:

- *Figure 1G: The authors indicate that LXR ligands and BH3-mimetics "were chosen based on prior studies". Given their goal is to identify an additive or synergic effect, it would be necessary to determine EC50 for each molecule before testing the combination of both. Authors would at least conduct this approach on 2 or 3 cell lines (for example one cell line of colon cancer and one of GBM).
- *Figure 1K-N: The definition of CI value (Combination Index) needs to be clarify in the legend. There is no statistical test to improve the robustness of these findings, in particular regarding results on A375 cell line.
- *Figure 2F: Authors should present results with histograms that provide statistical information's and facilitate figure analysis for readers.
- *Figure 3A: Western blot against a LXR target, such as ABCA1 or ABCG1, should be included to monitored LXR623 effects.
- *Figure 7G-I: Authors indicate that "the combination treatment resulted in a marked lower cellular density accompanied by significantly more cell death in the form of necrosis as well as apoptosis and a reduction in mitotic figures (Figure 7G and 7H)" Quantification and statistic need to be added.
- *Suppl Figure 9: The use of shNR1H2 is not sufficient to conclude, the authors need to show that in the presence of shNR1H2, the effect of LXR623 is abolished.
- *As stated above, TGCA datasets exploration will increase the rational of these findings.

Referee #3 (Remarks for Author):

In this manuscript, the authors demonstrate a synergy of LXR agonists and BH-3 mimetics for targeting solid tumors, including GBMs, based on unanticipated metabolic reprogramming of oxidative phosphorylation and central carbon metabolism. The use a variety of model systems in vitro and in vivo, and perform a series of epistatic experiments to demonstrate the synergy. The strengths of the paper include the importance of the topic, the epistatic experiments that show a causal pathway, and the potential importance of the implications. The data are compelling and the experiments are well controlled. The paper will be a valuable contribution to the literature.

The following suggestions are made to enhance the readability of the paper. The following issues are raised, which if addressed, would greatly strengthen the paper:

- 1) Fig 1 launches into the the BCL-2 family members indicating priming, but the data are buried in a GSEA plot in the supplement. Showing that more clearly and more explicitly will help readers understand this critical point.
- 2) Fig 1 K-N are synergy plots, but the axes are unlabeled and most readers will not understand the plots.
- 3) The link to ATF4 makes great sense, but no context is provided for the readers. It would be very helpful to introduce why they examined ATF4, before showing the data about it.
- 4) Fig 5 panel O, are very nice data, but they are not explained to readers in the figure legend. More explanation is needed to make this figure accessible.

Author Correspondence

28th June 2019

Thank you for the earlier correspondence regarding our manuscript.

We feel delighted that you gave us the opportunity to resubmit our manuscript. We have started the revision process and in this context I wanted to ask you a question related to the revision.

“Particular attention should be given to increasing the clinical relevance of the study, by adding clinical data and/or data from a transgenic mouse model.”

We have now interrogated the TCGA database and have found the correlation described by reviewer 1 and will be able to provide significant and novel data in this context (increasing the clinical relevance). Regarding the transgenic mouse model, we were wondering whether it was noted that we had already a patient-derived xenograft model (GBM43) included as part of our previous figure 7 (in vivo studies with the drug combinations). In addition, we had a short term GBM PDX culture in our figure 1. Therefore, I was wondering whether under these circumstances we would still need a transgenic or alternative in vivo model given the presence of a PDX model (which are considered to be the most state of the art systems to study therapies in preclinical cancer research) and our additional clinical data?

Editorial Correspondence

4th May 2019

I have now heard back from the first referee, who states:

"As I suggest in my previous review, transgenic model is not mandatory. If the authors get the opportunity to do it, this is a plus. Alternatively, PDX is OK. This is less relevant regarding physiopathology as it remains a subcutaneous injection and did not recapitulate microenvironment of the tumor. Besides, I understand that in vivo model is not easy to manage. Analyses done on TCGA sounds good. Thus, together with PDX, it is acceptable for me."

Point-by-point responses to reviewers' comments

We thank the Editors and reviewers for the appreciation of our work and the thorough review and insightful comments, which have significantly helped to provide additional perspectives on the presented work. We have taken these comments very seriously and have performed additional experiments to improve the paper in accordance with the reviewers' recommendations.

Referee #2 (Comments on Novelty/Model System for Author):

This study is of interest for medical questions surrounding cancer treatment. However, given that biological modeling is restricted to cancer cell lines, the clinical relevance is limited.

Referee #2 (Remarks for Author):

Nguyen T. et al. report inhibitory responses to LXRb activation and BH-3 mimetics on growth of various solid tumor derived cell lines including colon carcinoma, melanoma and glioblastoma. The authors conducted a molecular investigation to decipher how LXR ligands leads to downregulate OXPHOS pathway in tumor cells. They provide strong evidence that these effects are sustained by the upregulation of NOXA in an ATF4-dependent fashion and impact on the energy balance. In consequence TCA cycle is strongly downregulated that sensitive cells to BH3-mimetics. Technical approaches are relevant and support conclusions drawn. Research is well conducted. Overall, proposed article provides relevant evidences that sustain the synergistic effects of both targeting LXR signaling and Bcl-xL family in order to induce apoptosis of tumor cells. The focus on OXPHOS pathway in cancer show a growing interest in the field. Cancer cell sensitivity recurrence to chemotherapeutic agent using mitochondrial processes is a hot topic. Thus, this article represents a significant contribution.

The main weakness of this study come from that biological modeling is restricted to cell line using. Article will gain impact with transgenic mouse model (APCmin for colon cancer for example), as well as clinical data that could support GSEA results obtained on cell culture experiments. Especially, could we observe a correlation between the cluster of LXR target genes and mitochondrial-related gene expression described in Suppl Figure 1C-E in patient datasets.

We thank the reviewer for her/his careful assessment and appreciation of our manuscript. As requested, we have included data from the TCGA to enhance the clinical relevance. In addition, our in vivo and in vitro data contains patient-derived xenograft models.

Minor comments:

*Figures and Suppl Figures numeration is heterogeneous. Figure call needs to be in the correct order. Figure 1A 1C and 1B 1D needs to be Figure 1A 1B and 1C 1D. Similarly, panel number needs to be update for Suppl Figure 1, 5, 6 and 7 regarding the manuscript text apparition.

We fully agree with the reviewer. We relabeled the figures numeration so that they appear in the correct order.

*The section untitled "Activation of LXRb induces energy deprivation by shutting..." is too long, more than 8 pages! This part of the manuscript needs to be shortened as well as separated in various sections with only one message for each of them.

We shortened and divided this section in smaller paragraph accordingly. We agree with the reviewer that this has improved the readability and flow of the manuscript – Manuscript page 8.

* Given that authors claimed "These results nicely reinforce our in vitro results and raise hope that such a combination treatment might have implications for patients given the fact that both compounds have been tested in clinical trials", the following reference missing: Katz A, et al. Safety, pharmacokinetics, and pharmacodynamics of single doses of LXR-623, a novel liver X-receptor agonist, in healthy participants. *J Clin Pharmacol.* 2009 Jun;49(6):643-9. doi: 10.1177/0091270009335768. Authors should soften this comment regarding LXR623 exhibit significant side effects.

We added the reference in the text according to the reviewer's comments and softened the above statement accordingly – Manuscript page 22.

*The use of 20µM LXR623 is questionable as we could expected to be out of the range of LXR specific effects.

Although the concentration appears to be high this is likely related to the fact that in the presence of full media LXR623 mediated cell killing is attenuated due to the abundance of LDL-cholesterol in conventional medium. In experiments with lipoprotein-deficient serum LXR623 is more efficient to induce its effects.

Major comments:

*Figure 1G: The authors indicate that LXR ligands and BH3-mimetics "were chosen based on prior studies". Given their goal is to identify an additive or synergic effect, it would be necessary to determine EC₅₀ for each molecule before testing the combination of both. Authors would at least conduct this approach on 2 or 3 cell lines (for example one cell line of colon cancer and one of GBM).

We thank the reviewer and clarified this critical point. We analyzed the cell viability following treatment with LXR623 and ABT263 in the different cell lines and determined the EC₅₀ of each drug as recommended by reviewer. In order to calculate whether or not a combination treatment is synergistic, additive or antagonistic it is necessary to have adequate dose responses. To calculate the CI values (Combination index), we adhered to the approach described by Chou (Cancer Res. 2010 Jan 15;70(2):440-6. doi: 10.1158/0008-5472.CAN-09-1947) (Manuscript page 5). In turn, we included the normalized isobolograms as part of figure 1K-N and Appendix Figure 1A-E. Accordingly, we have extended the statement in the manuscript and explain that we performed dose responses and calculated the synergy based on the "median-effect equation". The related citation is provided as well (Manuscript page 5).

"*Figure 1K-N: The definition of CI value (Combination Index) needs to be clarify in the legend. There is no statistical test to improve the robustness of these findings, in particular regarding results on A375 cell line."

We defined the CI value in the legend by stating that the CI value was derived by the methodology described by Chou (see above – Manuscript page 5 and Manuscript page 37). More detailed explanation about this experiment was implemented in the Material and Method section (Manuscript page 28). To enhance the robustness, we calculated the mean and SEM for each of these experiments to provide a better idea about the robustness of these findings (Figure 1K-N). We have provided the normalized isobologram based on the dose-responses of the single treatments.

*Figure 2F: Authors should present results with histograms that provide statistical information's and facilitate figure analysis for readers.

As requested, we have provided statistical information of the experiment in figure 2F. These quantifications are now part of Figure EV3 (expanded view figure).

*Figure 3A: Western blot against a LXR target, such as ABCA1 or ABCG1, should be included to monitored LXR623 effects.

We fully agree with the reviewer's assessment. We assessed the ABCA1 protein level in response to increasing concentrations of LXR623. As anticipated, we detected an increase of ABCA1 protein level following LXR623 treatment in different cell lines. These results are now part of Figure 3B.

*Figure 7G-I: Authors indicate that "the combination treatment resulted in a marked lower cellular density accompanied by significantly more cell death in the form of necrosis as well as apoptosis and a reduction in mitotic figures (Figure 7G and 7H)" Quantification and statistic need to be added. The statistical information of Figure 7G-I was analyzed and added to show the combination treatment induced more cell death in the form of necrosis as well as apoptosis – Figure 7J-L.

*Suppl Figure 9: The use of shNR1H2 is not sufficient to conclude, the authors need to show that in the presence of shNR1H2, the effect of LXR623 is abolished.

As requested by the reviewer, we have tested the efficacy of LXR623 in the context of silencing experiments. As anticipated, we found that silencing of LXRb attenuates LXR623 driven reduction of oxygen consumption rate and ATP production in HCT116 as well. The data have been integrated as part of Appendix Figure 9E-G.

*As stated above, TGCA datasets exploration will increase the rational of these findings.

As requested by the expert reviewer, we have interrogated the TGCA datasets in connection with our findings. – Figure 1F and Expanded View Figure 1G-K. We found a correlation between the

effects of LXR623 on mitochondrial metabolism and the TGCA datasets. The LXR623 treatment induced the ABCA1 protein level that is accompanied by a reduction of mitochondrial translation/transcription and reduction in mitochondrial protein complexes in HCT116 colonic carcinoma cells. Similarly, in colorectal carcinoma patients, high levels of ABCA1 correlated with reduced levels of genes related to mitochondrial translation/transcription and mitochondrial protein complexes (Figure 1 and Expanded View Figure 1-2). Moreover, patients with high ABCA1 levels displayed highly significantly reduced transcript levels of OXPHOS related mRNA (Volcano plot – Figure 1F), which is in line with the effect of LXR agonists on mitochondrial energy metabolism (Figure 1 and Expanded View Figure 1-2). We concluded with pathway analysis of the most significant down-regulated genes in patients with high vs. low levels of ABCA1, which unequivocally demonstrated suppression of mitochondrial metabolism (Expanded View Figure 2B).

Referee #3 (Remarks for Author):

In this manuscript, the authors demonstrate a synergy of LXR agonists and BH-3 mimetics for targeting solid tumors, including GBMs, based on unanticipated metabolic reprogramming of oxidative phosphorylation and central carbon metabolism. The use a variety of model systems in vitro and in vivo, and perform a series of epistatic experiments to demonstrate the synergy. The strengths of the paper include the importance of the topic, the epistatic experiments that show a causal pathway, and the potential importance of the implications. The data are compelling and the experiments are well controlled. The paper will be a valuable contribution to the literature.

The following suggestions are made to enhance the readability of the paper. The following issues are raised, which if addressed, would greatly strengthen the paper:

We thank the reviewer for the appreciation of our work.

1) Fig 1 launches into the the BCL-2 family members indicating priming, but the data are buried in a GSEA plot in the supplement. Showing that more clearly and more explicitly will help readers understand this critical point.

As recommended by the reviewer, we relocated this figure to main panel to stress this critical point (Figure 1D).

2) Fig 1 K-N are synergy plots, but the axes are unlabeled and most readers will not understand the plots.

The axes were labeled to make the reader to understand the plots (Appendix Figure 1A-D). In addition, we implemented the normalized isobolograms related to the combination treatments and dose-escalations (Figure 1K-N). Moreover, we included more information on the methodology how to calculate drug synergism based on the method by Chou (Material and Methods section page 24).

3) The link to ATF4 makes great sense, but no context is provided for the readers. It would be very helpful to introduce why they examined ATF4, before showing the data about it.

In our study, we found that LXR activation inhibits oxidative phosphorylation with a pronounced decline in ATP levels and activation of the integrated stress response with upregulation of ATF4. ATF4 drives Noxa expression, leading to sensitization to cell death induction by the clinically validated Bcl-xL inhibitor ABT263. We provided more information why we focused on ATF4 in the context of our mechanism studies - Manuscript Page 9.

4) Fig 5 panel O, are very nice data, but they are not explained to readers in the figure legend. More explanation is needed to make this figure accessible.

We added additional text to the legend of Fig 5O to explain more the graphical depiction of glucose carbon tracing in TCA cycle as reviewer's recommendation (Figure legend 5O - Manuscript page 40).

2nd Editorial Decision

30th July 2019

Thank you for the submission of your revised manuscript to EMBO Molecular Medicine, and please accept my apologies for the delay in getting back to you, due to my recent traveling. We have now heard back from the two referees who were asked to reassess your work. As you will see the reviewers are now supportive of publication, and I am pleased to inform you that we will be able to accept your manuscript pending minor editorial amendments and a response to the minor comments from referee #3.

REFEREE REPORTS

Referee #2 (Remarks for Author):

Globally, authors answered critical requests made during reviewing. The paper is now suitable for publication.

Referee #3 (Remarks for Author):

The authors have done a commendable job of addressing the comments. the new data strengthen the paper, and it will be a valuable contribution to the literature. One minor issue in the revision relates to the new Fig 1F. the data are clear, but the presentation is not. I would suggest that the figure legend for 1F be re-written to explain the axes and what the data are showing. Without doing this, readers will have trouble understanding these new data.

2nd Revision - authors' response

1st August 2019

Referee #2 (Remarks for Author):

Globally, authors answered critical requests made during reviewing. The paper is now suitable for publication.

We thank the reviewer for her/his appreciation of our manuscript.

Referee #3 (Remarks for Author):

The authors have done a commendable job of addressing the comments. the new data strengthen the paper, and it will be a valuable contribution to the literature. One minor issue in the revision relates to the new Fig 1F. the data are clear, but the presentation is not. I would suggest that the figure legend for 1F be re-written to explain the axes and what the data are showing. Without doing this, readers will have trouble understanding these new data.

We thank the reviewer for her/his recommendation. We explained in more detail the axes of figure legend 1F to improve the clarity of the data.

Corresponding Author Name: Markus D. Siegelin

Journal Submitted to: Embo Molecular Medicine

Manuscript Number: EMM-2019-10769